# On the Role of General Function Approximation in Offline Reinforcement Learning

**Chenjie Mao**[1 2]**, Qiaosheng Zhang**[1]**, Zhen Wang**[3]**, Xuelong Li**[4]

[1]Shanghai Artificial Intelligence Laboratory, [2]Huazhong University of Science and Technology
[3]Northwestern Polytechnical University
[4]Institute of Artificial Intelligence (TeleAI), China Telecom Corp Ltd

## Abstract

We study offline reinforcement learning (RL) with general function approximation. General function approximation is a powerful tool for algorithm design and analysis, but its adaptation to offline RL encounters several challenges due to varying approximation targets and assumptions that blur the real meanings of function assumptions. In this paper, we try to formulate and clarify the treatment of general function approximation in offline RL in two aspects: (1) analyzing different types of assumptions and their practical usage, and (2) understanding its role as a restriction on underlying MDPs from information-theoretic perspectives. Additionally, we introduce a new insight for lower bound establishing: one can exploit model-realizability to establish general-purpose lower bounds that can be generalized into other functions. Building upon this insight, we propose two generic lower bounds that contribute to a better understanding of offline RL with general function approximation.

## 1 Introduction

*Reinforcement Learning* (RL; Sutton & Barto (2018)) studies learning optimal policies by interacting with the environment. Thanks to the increase in computation powers and the development of powerful function approximators, this *learning-based control* methodology has had tremendous success in recent years, as seen in studies such as Mnih et al. (2013); Silver et al. (2016); Kalashnikov et al. (2018). However, despite the significant progress and its great potential, RL algorithms face challenges in real-world applications like Healthcare (Gottesman et al., 2018; 2019; Tang et al., 2023) and Autonomous Driving (Kiran et al., 2021), due to the risk, cost, or ethical concerns associated with online explorations. A promising idea for applying RL to these domains is to utilize the dataset collected in advance to learn policies—this line of research falls into the area of *offline RL* (also known as batch RL; Lange et al. (2012); Levine et al. (2020)).

Theoretical studies of offline RL range from exceptional cases such as tabular Markov decision processes (MDPs; Wang et al. (2022); Rashidinejad et al. (2021); Xie et al. (2021b); Yin & Wang (2021); Li et al. (2022); Yan et al. (2022)) and low-rank MDPs (Jin et al., 2020; Xie et al., 2021a; Zanette et al., 2021; Yin et al., 2022; Huang et al., 2023) to the broader and more general setting of function approximation (Chen & Jiang, 2019; Liu et al., 2020; Xie & Jiang, 2020a; Foster et al., 2021b; Xie et al., 2021a; Cheng et al., 2022; Zhan et al., 2022). As algorithms and accompanying theoretical results for tabular MDPs and low-rank MDPs are incompatible with complex real-world problems attributable to the enormous states/actions or the complicated dynamics, general function approximation is often favored.

From the perspective of practical applications, general function approximation also enjoys its own significance. It characterizes how the approximation power and the complexity of approximators (e.g., support vector machine (Cortes & Vapnik, 1995) and neural net (Goodfellow et al., 2016)) jointly affect the performance of learning algorithms. In the analysis of offline RL, the approximation powers of function classes can be categorized into two types (Chen & Jiang, 2022; Zhan et al., 2022; Ozdaglar et al., 2022; Mao, 2023): *realizability-type* and *completeness-type*. Given a function class $\mathcal{F}$ and an approximation target $\mathcal{F}^{\star}$, the assumption $\mathcal{F}^{\star} \subseteq \mathcal{F}$ is considered realizability-type ($\mathcal{F}$ is

---

Corresponding authors: Zhen Wang (w-zhen@nwpu.edu.cn) and Xuelong Li (xuelong_li@ieee.org). Xuelong Li is the Chief Technology Officer (CTO) and the Chief Scientist of the China Telecom.

then called *realizable*) if $|\mathcal{F}^\star| = 1$ (e.g., $\mathcal{F}^\star = \{Q^\star\}$), and is considered completeness-type if there exists a (known) bijection between $\mathcal{F}^\star$ and another realizable function class $\mathcal{G}$. Most theoretical algorithms and analyses assume an exponentially large function class to achieve realizability-type assumptions and provide sample-complexity guarantees that are polynomial in $\log(|\mathcal{F}|)$. As a consequence, completeness-type assumptions can lead to an exponential performance bound since the approximation target itself may be exponentially large. Moreover, one of the most common completeness-type assumptions is *Bellman-completeness* (the value function class is closed under Bellman backup), where two function classes (e.g., $\mathcal{F}$ and $\mathcal{G}$ above) are identical. This form of "self-completeness" is more stringent because the performance guarantee may be violated by just adding one function into the function class (in contrast to the general intuition in supervised learning (Chen & Jiang, 2019)). Thus, in most cases, realizability-type assumptions are preferred.

The assumptions of functions are also impacted by the quality of the dataset, which forms another aspect of learnability in offline RL. On one hand, Xie & Jiang (2020a) shows that realizability-type assumptions (concretely, $Q^\star$-realizability) combined with data assumptions that are even stronger than the classic and stringent *exploratory coverage* (Precup et al. (2000); Antos et al. (2006); see also Section 2.2), are enough for learning a near-optimal policy.[2] On the other hand, there are works (e.g., Liu et al. (2020); Xie et al. (2021a); Rashidinejad et al. (2022); Zhu et al. (2023)) showing that with some mild completeness-type assumptions, even a dataset with partial coverage is enough for learning a good policy. However, despite numerous efforts devoted, no work has yet achieved learnability under both weak assumptions on function classes and datasets. This raises the question: what are the limitations of general function approximation in offline RL?

In learning theory, the fundamental limitations of problems are often identified by the minimax lower bound. However, establishing such lower bounds becomes a challenging task when dealing with general function approximation in offline RL. This is primarily due to the significant variation in the functions that we want to approximate and the relationships between them. What is even worse is that establishing lower bounds for certain properties does not necessarily imply a fundamental barrier to learnability. For instance, if a lower bound is established for value functions, it does not necessarily imply that the problem cannot be learned. Additional assumptions on properties such as the *density ratio* (Liu et al., 2018; Uehara et al., 2019) may make it learnable.

This paper enhances the understanding of general function approximation in offline RL in the following aspects:

1. (Section 3) Following the insight from Chen & Jiang (2022); Zhan et al. (2022); Ozdaglar et al. (2022); Mao (2023), we classify function assumptions in offline RL into completeness-type and realizability-type. Based on this taxonomy, we analyze their practical usage and demonstrate that completeness-type assumptions are often necessary to approximate targets for every possible policy in algorithms.

2. (Section 4) We show that function classes in RL can be viewed as a restriction on possible MDPs. We can concretize this restriction as *model-realizability*, i.e., the assumption that we have a MDP class $\mathcal{M}$ containing the real MDP. This allows us to establish lower bounds for model-realizability and extend them to other function classes.

3. (Section 5) We propose a generic lower bound in Theorem 1 for general function approximation. Based on the principle of lower bound construction presented in Section 4, we derive some interesting corollaries from Theorem 1:[3]

  (a) Informal version of Corollary 1: Given realizability-type assumptions on the value function and the density ratio for a specific policy within a policy class, along with "any" data-coverage assumption, we cannot learn a better policy than the aforementioned one.

  (b) Informal version of Corollary 2: Given *exploratory-accurate* realizability-type assumptions on any functions that take the state space as input for a specific policy within a policy class, along with "any" data-coverage assumption, we cannot learn a better policy than the aforementioned one.

---

[2]Their data assumptions require that (i) the state margin of the data distribution both scales with the transition kernel and scales with the initial distribution, and (ii) for every state, the behavior policy (i.e., the conditional distribution of the action given that state, which is derived from the data distribution) should cover all actions.

[3]See Section 3 for the definitions of *exploratory-accuracy* and *behaviour-accuracy*.

(c) Informal version of Corollary 3: Given *behaviour-accurate* realizability-type assumptions on any functions for a specific policy within a policy class, along with "any" data-coverage assumption, we cannot learn a better policy than the aforementioned one.

4. (Section 6) We augment results from Section 5 with $Q^\star$-realizability at the cost of introducing partial coverage (Theorem 2). A limitation of this augmented lower bound is that the covered policy is not optimal.

## 2 PRELIMINARIES AND RELATED WORKS

This section introduces basic settings and concepts in offline RL. See Table 1 in Appendix A for the full list of notations.

### 2.1 MARKOV DECISION PROCESSES (MDPS)

This paper studies infinite-horizon discounted Markov Decision Processes (abbreviated as MDPs for short). An MDP is characterized by a tuple $(\mathcal{S}, \mathcal{A}, P, R, \mu_0, \gamma)$, where $\mathcal{S}$ is the state space, $\mathcal{A}$ is the action space, $P : \mathcal{S} \times \mathcal{A} \to \Delta(\mathcal{S})$ is the transition kernel, $R : \mathcal{S} \times \mathcal{A} \to [0, R_{\max}]$ is the (deterministic) reward function,[4] $\mu_0 \in \Delta(\mathcal{S})$ is the initial state distribution, and $\gamma \in [0, 1)$ is the discount rate. We use $\nu$ to denote the uniform measure of $\mathcal{A}, \mathcal{S}$, or $\mathcal{S} \times \mathcal{A}$, depending on the context. We will add a subscript $M$ to a certain symbol (e.g., $P_M$) when we want to highlight the dependency with a specific MDP $M$. A policy $\pi : \mathcal{S} \to \Delta(\mathcal{A})$ is a mapping from the state space to a distribution on the action space,[5] and with a slight abuse of notation, we sometimes use $\pi$ to denote deterministic policies. We say a policy $\pi$ induces a (random) *trajectory* $\{s_0, a_0, r_0, s_1, a_1, r_1, \ldots, s_i, a_i, r_i, s_{i+1}, \ldots\}$ if $s_0 \sim \mu_0, a_i \sim \pi(\cdot|s_i), r_i = R(s_i, a_i), s_{i+1} \sim P(\cdot|s_i, a_i)$ for all $i \in \mathbb{N}$. For a policy $\pi$, we define its *value function* as the expected return from a specific state or a state-action pair, i.e., $V_\pi(s) := \mathbb{E}[\sum_{i=0}^\infty \gamma^i r_i | s_0 = s, a_i \sim \pi(\cdot|s_i)]$ and $Q_\pi(s, a) := \mathbb{E}[\sum_{i=0}^\infty \gamma^i r_i | s_0 = s, a_0 = a, a_i \sim \pi(\cdot|s_i)]$. Since $r_i \in [0, R_{\max}]$, the value function is upper bound by $V_{\max} := R_{\max}/(1 - \gamma)$. We denote $Q^\star$ as the unique solution of the celebrated Bellman-optimality equation and name it as *optimal value function*. The state version of $Q^\star$ is defined as $V^\star(s) := \max_{a \in \mathcal{A}} Q^\star(s, a)$. We assess the performance of $\pi$ through the expected return from the initial state $s_0$ that follows from $\mu_0$: $J(\pi) := \mathbb{E}_{s \sim \mu_0}[V_\pi(s)]$. We also define $J^\star := \max_\pi J(\pi)$ as the optimal return, and take a policy $\pi$ as an optimal policy if $J(\pi) = J^\star$. We say that a policy is *optimal everywhere* if its state value function is $V^\star$, and such a policy is denoted as $\pi_e^\star$. For a policy $\pi$ and a transition kernel $P$, we define the *state transition kernel* $P_\pi : \mathcal{S} \to \Delta(\mathcal{S})$ as $P_\pi(s'|s) := \int_{\mathcal{A}} P(s'|s, a)\pi(a|s)d\nu(a)$. An $\mathcal{S}$-*trajectory* of a policy $\pi$ is a sequence $\{s_0, r_0, s_1, r_1, \ldots, s_i, r_i, s_{i+1}, \ldots\}$ such that $s_0 \sim \mu_0, r_i = \mathbb{E}_{a \sim \pi(\cdot|s_i)}[R(s_i, a)], s_{i+1} \sim P_\pi(\cdot|s_i)$ for all $i \in \mathbb{N}$. The induced distribution on the state-action pair of a policy $\pi$ is defined as $d_\pi(s, a) := (1 - \gamma)\left[\sum_{i=0}^\infty \gamma^i \mathbb{P}(s_i = s, a_i = a | s_0 \sim \mu_0, \pi)\right]$. In certain cases, we may come across trajectories and induced distributions that do not originate from $\mu_0$. However, unless specified otherwise, we default to using $\mu_0$ as the initial distribution.

### 2.2 OFFLINE POLICY LEARNING

In the general framework of offline policy learning, we are given a dataset $\mathcal{D}$ consisting of $N$ i.i.d. $(s, a, r, s')$ tuples. Given a *data distribution* $d^\mathcal{D}$, the dataset $\mathcal{D}$ is collected such that $(s, a) \sim d^\mathcal{D}, r = R(s, a)$ and $s' \sim P(\cdot|s, a)$. A learning algorithm, that is denoted as $\mathfrak{A}$, is a mapping from the dataset $\mathcal{D}$ to a policy $\widehat{\pi}$.

Difficulties in offline policy learning arise from limited access to pre-collected datasets, with no further interaction with the environment permitted. The representability of the dataset, which determines the fundamental limit of algorithms, is often evaluated through the *concentration coefficient* (Munos, 2003; 2005). The concentration coefficient measures the distribution shift from the data distribution $d^\mathcal{D}$ to the induced distribution of some policies. Concretely, the classical *exploratory coverage* assumption assumes that there exists a constant $C_{\exp}$ such that for any (sometimes also non-stationary) policy $\pi$, $C_{\exp} \geq \left\|\frac{d_\pi}{d^\mathcal{D}}\right\|_\infty$. The *partial coverage* assumption (Liu et al., 2020; Xie et al., 2021a), on the other hand, assumes that there exists a constant $C_{\mathrm{par}}$ such that for some policy $\pi$, $C_{\mathrm{par}} \geq \left\|\frac{d_\pi}{d^\mathcal{D}}\right\|_\infty$.

---

[4]Stochastic rewards contain deterministic rewards. Thus, our lower bound can also be applied to the stochastic reward setting.

[5]This paper primarily focuses on stationary policies. *Non-stationary* policies, which can change over different steps, are denoted as $\pi_{\mathrm{non}}$.

The learning goal (i.e., the criterion of learnability) in offline RL is to obtain a policy that is at least no worse than the covered one (i.e., if $\pi$ is covered, $J(\widehat{\pi}) \geq J(\pi)$ asymptotically).

## 2.3 GENERAL FUNCTION APPROXIMATION

General function approximation in RL is a broader framework that extends beyond tabular MDPs or low-rank MDPs. It is assumed that we have some function classes with good properties (e.g., containing the approximation target) while still maintaining limited complexity (measured by cardinality, metric entropy, etc.). Most works allow function classes to have exponential cardinalities or covering numbers. The complexity of function classes is determined by the complexity of the underlying MDPs. Thus, these modeling assumptions can also be viewed as implicit structural assumptions for MDPs. Further, when considering algorithm designs and analyses in tabular or low-rank settings, careful investigations suggest that the structures of MDPs are influential because of their role in function classes modeling (e.g., modeling state-action value functions or transition kernels) and some structure-specific quantities (e.g., the feature covariance matrix). In particular, generalizing these structural-specific quantities to complex real-world problems is much more challenging. Thus, general function approximation is crucial when considering algorithm design in RL.

From a historical perspective, the analyses of offline RL with general function approximation are extended from approximate dynamic programming (ADP; Bertsekas & Tsitsiklis (1996); Munos (2003; 2005)). Works in the early stages mainly focus on the error propagation bounds in the process of dynamic programming. For instance, in policy iteration, how errors from value estimation in each step affect the finally approximated optimal value function. Later on, the complexity of the function classes being used (mainly the value function class) and non-asymptotic analyses for the dataset are taken into consideration (Szepesvari & Munos, 2005; Antos et al., 2006; 2007; Munos & Szepesvari, 2008). Through the development, concepts like concentration coefficient and Bellman-completeness are proposed, and strong assumptions on both function classes and datasets are considered necessary for learning near-optimal policies (Chen & Jiang, 2019). The lack of understanding of the fundamental requirement of offline RL was first noticed long later by Chen & Jiang (2019), and it asks if learning under weaker assumptions is possible for offline RL. From then on, there have been emerging works proposing PAC-learnable algorithms under either mitigated assumptions on dataset (Jiang & Huang, 2020; Liu et al., 2020; Xie & Jiang, 2020a; Uehara & Sun, 2021; Xie et al., 2021a; Cheng et al., 2022; Rashidinejad et al., 2022; Zhu et al., 2023) or on function classes (Xie & Jiang, 2020a). However, despite these advancements, comprehension of general function approximation (e.g., how different assumptions interact and play their roles) is still lacking.

## 2.4 LOWER BOUNDS IN OFFLINE RL

While information-theoretic lower bounds have been extensively studied in both tabular MDPs (Rashidinejad et al., 2021; Xie et al., 2021b; Li et al., 2022) and linear MDPs (Jin et al., 2020; Wang et al., 2020; Amortila et al., 2020; Zanette, 2020; Chen et al., 2021; Wang et al., 2021), lower bounds considering general function approximation in offline RL are still lacking. The most notable work is Foster et al. (2021a), demonstrating that learning is impossible without Bellman-completeness, even in the presence of an exploratory dataset. However, their lower bounds are limited since they only consider value functions and are value-based, while recent years have seen the significant development of newly proposed properties such as the *density ratio*. The complexity and difficulty of analyses in general function approximation explode for the increasing number of properties we can model. Moreover, the fundamental requirement of learnability in general function approximation—which properties we should approximate and what assumptions we should demand—is still unclear.

## 3 ON THE PRACTICAL USAGE OF GENERAL FUNCTION APPROXIMATION

Recall that in the analysis of general function approximation, the assumptions can be categorized into two types: realizability-type and completeness-type (Chen & Jiang, 2022; Zhan et al., 2022; Ozdaglar et al., 2022; Mao, 2023). Since the gap between these two types of assumption is sometimes vague in previous works, we first formalize our discussion with some definitions.

**Definition 1** (Realizability-type assumption). Given a function class $\mathcal{F}$ and a target function class $\mathcal{F}^\star$, we say that the assumption $\mathcal{F}^\star \subseteq \mathcal{F}$ is *realizability-type* if $|\mathcal{F}^\star| = 1$ (we can thus denote $\mathcal{F}^\star = \{f^\star\}$), and say that $\mathcal{F}$ is realizable under this assumption.

Additionally, an *exploratory-accurate* realizability-type assumption assumes that $\mathcal{F}^\star$ is only accurate under the induced distribution of any non-stationary policy, i.e., $\forall \pi_{\text{non}}, \min_{f \in \mathcal{F}} \|f - f^\star\|_{\infty, d_{\pi_{\text{non}}}} = 0$. A *behaviour-accurate* realizability-type assumption assumes that $\mathcal{F}^\star$ is only accurate under the induced distribution of a specific policy $\pi$, i.e., $\min_{f \in \mathcal{F}} \|f - f^\star\|_{\infty, d_\pi} = 0$.[6] In the above definition, $\|x\|_{p,q}$ denotes the $q$-weighted $L^p$ norm, i.e., $\|x\|_{p,q} := \left( \int x^p dq \right)^{1/p}$.

**Example 1** (Example of realizability-type assumptions). *We have a state-action value function class $\mathcal{Q} \subseteq (\mathcal{S} \times \mathcal{A} \to [0, V_{\max}])$ such that $\{Q^\star\} \subseteq \mathcal{Q}$. Corresponding to Definition 1, we thus have $\mathcal{F}^\star = \{Q^\star\}$ and $\mathcal{F} = \mathcal{Q}$.*

**Definition 2** (Completeness-type assumption). Given a function class $\mathcal{F}$ and a target function class $\mathcal{F}^\star$, we say that the assumption $\mathcal{F}^\star \subseteq \mathcal{F}$ is *completeness-type* if $|\mathcal{F}^\star| = |\mathcal{G}|$ for another realizable function class $\mathcal{G}$, and we say that $\mathcal{F}$ is complete with respect to (w.r.t.) $\mathcal{G}$ under this assumption.

**Example 2** (Example of completeness-type assumptions). *We have a policy class $\Pi \subseteq (\mathcal{S} \to \Delta(\mathcal{A}))$ and a state-action value function class $\mathcal{Q} \subseteq (\mathcal{S} \times \mathcal{A} \to [0, V_{\max}])$. We assume that $\pi^\star \in \Pi$. The assumption that $\{Q_\pi \mid \pi \in \Pi\} \subseteq \mathcal{Q}$ is completeness-type, and we say that $\mathcal{Q}$ is complete w.r.t. $\Pi$ under this assumption. Corresponding to Definition 2, we thus have $\mathcal{F}^\star = \{Q_\pi | \pi \in \Pi\}$, $\mathcal{F} = \mathcal{Q}$ and $\mathcal{G} = \Pi$.*

**Remark 1.** The completeness-type assumption is commonly used to ensure the existence of a function class that can minimize a set of loss functions indexed by another function class.

Completeness-type assumptions are common but may be harmful. Intuitively, making a completeness-type assumption is like making as many realizability-type assumptions as the number of functions in another realizable function class. This tiered structure would introduce a dilemma in the approximator design. Assuming that $\mathcal{F}$ is complete w.r.t. $\mathcal{G}$, one may wish $\mathcal{G}$ to be rich enough to capture the approximation target of itself. However, even adding just one function into $\mathcal{G}$ may break the completeness-type assumption on $\mathcal{F}$, rendering the guarantees associated with this assumption no longer applicable.

**Remark 2.** Although the richness of function classes would also affect the generalization error, the influence is *monotonic* in sample complexity and can be mitigated by large samples. This contrasts the completeness-type assumption since it can induce a constant performance gap for having just one more function in the function class (which is referred to as *non-monotonic*, therefore).

Moreover, constructing a function $\mathcal{F}$ that can model $|\mathcal{G}|$ targets is overly complicated and can spoil the final results. For a universal approximator without prior knowledge, assuming that we can approximate a single target (in cases of realizability-type assumptions) with an exponentially large function class (matching the classical $\log$ sample complexity bound), a complete function class would be "doubly-exponentially" large. In such a situation, the sample complexity, measured in terms of the polylogarithm of the cardinality of the complete function class, would become exponentially large. This renders it impractical and no longer meaningful.[7]

**Why do we need completeness-type assumptions** In RL, the primary goal is to learn a policy that meets certain criteria. We argue that most algorithms in RL can be considered "policy-based" and are designed with a specific set of policies in mind. For instance, value-based methods focus on updating policies that take actions with the highest value in each state for each value function, and model-based methods would consider optimal policies under the MDPs from the model class. Policies considered in these methods are pre-defined even before algorithms start. This means that most algorithms in RL can be regarded as those that take a policy class $\Pi$, a function class $\mathcal{F}$, and a set of assumptions $\mathbb{A}$ as input. Completeness-type assumptions are required to approximate targets well for each $\pi \in \Pi$.

**Example 3.** *The Fitted Q-iteration requires a state-action value function class satisfying Bellman-completeness, i.e., given a state-action value function class $\mathcal{Q} \subseteq (\mathcal{S} \times \mathcal{A} \to [0, V_{\max}])$, for each*

---

[6]In the case that $f$ is defined on $\mathcal{S}$ instead of $\mathcal{S} \times \mathcal{A}$, we can substitute $d_\pi$ with its state margin $\mu_\pi$.

[7]For example, the typical union bound would introduce logarithmic dependency in sample complexity, whichever function class we use. If we can approximate a target with a function class with cardinality $\exp(C)$, as the completeness-type assumption requires us to approximate $\exp(C)$ targets, the completeness-type assumption would require a function class with $\exp(\exp(C))$ elements. The sample complexity bound would be $\log(\exp(\exp(C))) = \exp(C)$.

$q \in \mathcal{Q}$, we have its Bellman update $\mathcal{T}q$ contained by $\mathcal{Q}$. It can be regarded as "policy-based" such that $\Pi := \{\pi_q | q \in \mathcal{Q}\}$, $\mathcal{F} = \mathcal{Q}$, and $\mathbb{A}$ assumes that there is a known mapping $\phi$ (maps from $q$ to $\pi_q$) such that for all $\pi \in \Pi$, $\mathcal{T}^\pi f \in \mathcal{F}$ for all $f \in \phi^{-1}(\pi)$.

In the above example, $\mathcal{T}$ denotes the Bellman operator, $\pi_q$ denotes the optimal policy for $q$, and $\mathcal{T}^\pi$ denotes the Bellman operator of policy $\pi$.

Two instances that contradict the aforementioned statements are model-realizability and the algorithm from Xie & Jiang (2020a). On one hand, while the upper bound for model-realizability has already been proposed (Uehara & Sun, 2021), model-realizability subsumes completeness-type assumptions (cf. Proposition 3). On the other hand, Xie & Jiang (2020a) use discretization to ensure good approximations for a refined $\mathcal{T}q$ for each $q$, and they require a dataset with quite strong coverage assumptions that are a bit unrealistic.

In summary, we want to ask:

**Question 1.** *Can we learn good policies without approximating targets well for all $\pi \in \Pi$?*

To provide a more concise presentation, we will first introduce insights and tools related to learnability and general function approximation in the next section. This will provide the necessary background for answering Question 1.

## 4 ANATOMY OF GENERAL FUNCTION APPROXIMATION APROPOS OF LEARNABILITY

This section provides a detailed analysis of general function approximation in offline RL with a focus on learnability.

**On the role of function classes** A reinforcement learning problem can be defined by an MDP $M$, supplemented by additional information in the form of a function class $\mathcal{F}$ and a set of assumptions $\mathbb{A}$. $\mathcal{F}$ is allowed to encompass multiple function classes through cartesian products. From a theoretical perspective, it is crucial to understand the significance of $\mathcal{F}$ in the context of information-theoretic learnability. A key insight into this problem is given by the well-known definition of the minimax lower bound, which we denote as $\mathcal{L}$,

$$\mathcal{L}(\mathcal{F}, \mathbb{A}) := \min_{\mathfrak{A}} \max_{M \in \mathcal{M}(\mathcal{F}, \mathbb{A})} \mathcal{C}(\mathfrak{A}, M), \tag{1}$$

where $\mathcal{C}$ is a real-valued objective function we can ask algorithms to minimize, and $\mathcal{M}(\mathcal{F}, \mathbb{A})$ enumerates all possible MDPs with $\mathbb{A}$ satisfied w.r.t. $\mathcal{F}$ we may encounter.

The "max" part of Eqn. (1) can be understood as the presence of an adversary who selects the worst-case MDP from $\mathcal{M}(\mathcal{F}, \mathbb{A})$ for algorithms. Specifying to the concrete problem, there is a real model $M^\star$ that serves as the ground truth. Note that $M^\star \in \mathcal{M}(\mathcal{F}, \mathbb{A})$ by the definition of $\mathcal{M}(\mathcal{F}, \mathbb{A})$. In other words, The adversary can only select elements from $\mathcal{M}(\mathcal{F}, \mathbb{A})$ as the real model $M^\star$. To summarize, Eqn. (1) asserts implicitly that: function class $\mathcal{F}$ and assumption set $\mathbb{A}$ are a restriction on possible underlying MDPs that the adversary can select.

**Remark 3.** In offline RL, we are also given a dataset $\mathcal{D}$ with data distribution $d^\mathcal{D}$. With a slight abuse of notations, one can view $\mathcal{D}$ as a part of $\mathcal{F}$, $d^\mathcal{D}$ as a part of the MDP $M$, and the assumptions on the dataset $\mathcal{D}$ as a part of $\mathbb{A}$, respectively. These inclusions are only assumed in this section.

**Model-realizability: a principle for lower bound construction** Lower bounds for general function approximation are inherently specific to the type of function being considered. It is often believed that lower bounds derived for one type of function are not applicable to another. However, this is not the case for model-realizability. We argue that the lower bounds derived for model-realizability can also be applied to other types of functions. For any function class $\mathcal{F}$, we denote $\mathfrak{A}(\mathcal{F})$ as the set of all possible algorithms taking $\mathcal{F}$ as inputs. We denote $\mathcal{F}(\mathcal{M}, \mathbb{A})$ as a constructed function class from $\mathcal{M}$ and $\mathbb{A}$ such that $\mathbb{A}$ is satisfied for all $M \in \mathcal{M}$. It is assumed that the construction process is known. Proposition 1 below shows that lower bounds for a realizable $\mathcal{M}$ ($M^\star \in \mathcal{M}$) apply to $\mathcal{F}(\mathcal{M}, \mathbb{A})$ for any $\mathbb{A}$.

**Proposition 1.** *Algorithms taking $\mathcal{F}(\mathcal{M}, \mathbb{A})$ as inputs must perform no better than the best algorithm that takes the realizable $\mathcal{M}$ as input in the worst case:*

$$\min_{\mathfrak{A} \in \mathfrak{A}(\mathcal{F}(\mathcal{M}, \mathbb{A}))} \max_{M \in \mathcal{M}(\mathcal{F}(\mathcal{M}, \mathbb{A}), \mathbb{A})} \mathcal{C}(\mathfrak{A}, M) \geq \min_{\mathfrak{A} \in \mathfrak{A}(\mathcal{M})} \max_{M \in \mathcal{M}} \mathcal{C}(\mathfrak{A}, M). \tag{2}$$

Proposition 1 can be justified by the facts that: (i) the set of algorithms taking $\mathcal{F}(\mathcal{M}, \mathbb{A})$ as inputs is a subset of the set of algorithms that takes $M$ as inputs (we can transfer before applying algorithms); and (ii) the set $\mathcal{M}(\mathcal{F}(\mathcal{M}, \mathbb{A}), \mathbb{A})$ is a superset of the MDP class $\mathcal{M}$. The significance of Proposition 1 is that it provides a general principle for lower bound construction: one can first establish lower bounds based on the assumption of model-realizability, and then extend the bound to any function that may not have been accessible previously.

A more detailed discussion of how can we construct $\mathcal{F}$ from $\mathcal{M}$ and $\mathbb{A}$ is deferred to Appendix C due to the page limitation. Moreover, $\mathcal{F}$ may also depend on properties other than $\mathcal{M}$ (e.g., the value function of a policy). This type of functions is related to completeness-type assumptions. We also defer the elaboration of this to Appendix C.

**Remark 4.** It is worth noting that upper bounds for partial concentrability and model realizability have been established in previous works (e.g., Uehara & Sun (2021)). As a consequence, the lower bound we establish for $\mathcal{M}$ requires certain weakening of assumptions.

## 5 LOWER BOUNDS UNDER STRONG DATA ASSUMPTIONS

This section provides a negative answer to Question 1, even when the policy class $\Pi$ only contains two elements. We first interpret Question 1 through the goal presented below.

**Goal.** *Given a dataset $\mathcal{D}$, a policy class $\Pi$, a function class $\mathcal{F}$, and a set of assumptions $\mathbb{A}$, we wish to learn a policy $\widehat{\pi}$ such that $J(\widehat{\pi}) \geq J(\widetilde{\pi})$ for any policy $\widetilde{\pi}$, while only assuming that we can approximate certain targets for $\widetilde{\pi}$.*

This goal is closely related to the target of robust policy improvement (Cheng et al., 2022). Our data assumption is modified from Xie & Jiang (2020a), and is stronger than classic partial coverage or exploratory coverage assumptions, as well as the more refined data assumptions presented in recent years (Xie & Jiang, 2020a; Uehara et al., 2023). The primary lower bound is presented in Theorem 1. A detailed comparison with assumptions in previous works is deferred to Appendix F.

**Theorem 1.** *For any sample size $N$ and $\gamma \in [0, 1)$, there exist a family of MDPs $\mathcal{M}$ with the same state-action spaces, a reward function $R$, a transition kernel $\widetilde{P}$, a dataset $\mathcal{D}$ with distribution $d^{\mathcal{D}}$, a policy class $\Pi$, a state-action value function class $\mathcal{Q}$ and a state transition probability class $\mathcal{P}_{\mathcal{S}}$, such that*

1. *$\left\| \frac{d_{M, \pi_{\mathrm{non}}}}{d^{\mathcal{D}}} \right\|_{\infty} \leq 16$ for any (possibly non-stationary) policy $\pi_{\mathrm{non}}$ and for all $M \in \mathcal{M}$,*

2. *the behaviour policy $\pi_b$ (obtained from $d^{\mathcal{D}}$) satisfies $\pi_b(a \mid s) \geq 1/2$ for all $s \in \mathcal{S}$ and $a \in \mathcal{A}$,*

3. *the state margin $\mu^{\mathcal{D}}$ (obtained from $d^{\mathcal{D}}$) satisfies $P_M(s' \mid s, a)/\mu^{\mathcal{D}}(s') \leq 8$ for all $s \in \mathcal{S}, a \in \mathcal{A}, s' \in \mathcal{S}$ and $M \in \mathcal{M}$,* [8]

4. *$|\mathcal{A}| = |\mathcal{Q}| = |\mathcal{P}_{\mathcal{S}}| = |\Pi| = 2$,*

5. *$R$ is the reward function for all $M \in \mathcal{M}$,*

6. *there exists a (unknown) mapping $\xi \in (\mathcal{M} \to \Pi)$ such that for all $M \in \mathcal{M}$,*

    - *$\xi(M)$ has its state-action value function realized by $\mathcal{Q}$ (i.e., $Q_{M, \xi(M)} \in \mathcal{Q}$),*

    - *the state transition kernel of $\xi(M)$ is realized by $\mathcal{P}_{\mathcal{S}}$ with exploratory-accuracy, i.e., $\exists P_{\mathcal{S}} \in \mathcal{P}_{\mathcal{S}}$, such that for all (possibly non-stationary) policy $\pi_{\mathrm{non}}$, $\|P_{\mathcal{S}} - P_{M, \xi(M)}\|_{\infty, d_{\pi_{\mathrm{non}}}} = 0$,*

    - *$\widetilde{P}$ is partially accurate under the induced distribution of $\xi(M)$ (i.e., $\|\widetilde{P} - P_M\|_{\infty, d_{\xi(M)}} = 0$).*

*Any learning algorithm $\mathfrak{A}$—which takes $\mathcal{S}, \mathcal{A}, R, \mu_0, \gamma, \mathcal{D}, d^{\mathcal{D}}, \Pi, \mathcal{Q}, \widetilde{P}$ and $\mathcal{P}_{\mathcal{S}}$ as input and outputs a policy $\widehat{\pi}$—must satisfy*

$$\max_{M \in \mathcal{M}} \left( J_M(\xi(M)) - \mathbb{E}\big[J_M(\widehat{\pi})\big] \right) \geq \frac{\gamma^2}{8}.$$

---

[8] We define $0/0 = 1$.

*Moreover, for all $M \in \mathcal{M}$, we have that $J_M(\xi(M)) = J_M^\star$.*

**Remark 5.** While we are provided with the policy class $\Pi$, the learner has the freedom to select policies that do not belong to $\Pi$.

Our counterexample construction technique is inspired by the concept of "over-coverage" from Xie & Jiang (2020a); Foster et al. (2021b). Through the function transfering principle presented in Section 4, several interesting corollaries can be derived from Theorem 1. These corollaries are presented in Section 5.1 below.

We defer the proofs for this section to Appendix D.

## 5.1 INTERPRETATION OF THEOREM 1

To answer Question 1, we need to first clarify what are the targets of a policy.

**Definition 3.** *We say that a function mapping from $\mathcal{S} \times \mathcal{A}$ or $\mathcal{S}$ is $\pi$-related if for each entry of this function, we can calculate its value from the induced trajectory of $\pi$ from this entry. A function mapping from $\mathcal{S}$ is $(\pi, \mathcal{S})$-related if we can calculate its values from the induced $\mathcal{S}$-trajectory of $\pi$ from each entry.*

**Example 4.** *$Q_\pi$ is $\pi$-related since its value for each $(s, a) \in \mathcal{S} \times \mathcal{A}$ is the sum of the cumulative rewards of the trajectory thereby. Moreover, $V_\pi$ is $(\pi, \mathcal{S})$-related.*

First, as a response to marginalized importance sampling (MIS), we show in Corollary 1 that the realizability w.r.t. the density ratio is insufficient for learning a better policy. We denote the density ratio class as $\mathcal{W} \subseteq (\mathcal{S} \times \mathcal{A} \to [0, W_{\max}])$ where $W_{\max}$ is a constant, while the density ratio of a policy $\pi$ is defined as $w_\pi(s, a) := d_\pi(s, a)/d^{\mathcal{D}}(s, a)$.

**Corollary 1.** *Under the condition of Theorem 1, any algorithm that also takes $\mathcal{W}$, which has the density ratio realized ($\forall M \in \mathcal{M}, w_{\xi(M)} \in \mathcal{W}$) and only contains 4 elements, as input must have $\max_{M \in \mathcal{M}} \left(J_M(\xi(M)) - \mathbb{E}\left[J_M(\widehat{\pi})\right]\right) \geq \frac{\gamma^2}{8}$.*

The following corollary shows that modeling any function defined on $\mathcal{S}$ (as a consequence of $\mathcal{P}_{\mathcal{S}}$) with exploratory-accuracy is not enough for learning a better policy.

**Corollary 2.** *Under the condition of Theorem 1, for any function (which we denote as $f^\star$) we want to approximate that is $(\pi, \mathcal{S})$-related for a certain $\pi \in \Pi$ specified by $\xi$,[9] we can construct a function class $\mathcal{F}$ such that*

1. *$\mathcal{F}$ only contains four elements,*

2. *for all $M \in \mathcal{M}$, there exists $f \in \mathcal{F}$ such that for any (possibly non-stationary) policy $\pi_{\text{non}}$, $\|f - f_M^\star\|_{\infty, \mu_{\pi_{\text{non}}}} = 0$.*

*Algorithms from Theorem 1 that also take $\mathcal{F}$ as input must have $\max_{M \in \mathcal{M}} \left(J_M(\xi(M)) - \mathbb{E}\left[J_M(\widehat{\pi})\right]\right) \geq \frac{\gamma^2}{8}$.*

Furthermore, modeling any $\pi$-related function that is accurate under the induced distribution of $\pi$ (as a consequence of $\widetilde{P}$) is not enough for learning a better policy.

**Corollary 3.** *Under the condition of Theorem 1, for any function (which we denote as $f^\star$) we want to approximate that is $\pi$-related for a certain $\pi \in \Pi$ specified by $\xi$, we can construct a function class $\mathcal{F}$ such that*

1. *$\mathcal{F}$ only contains two elements,*

2. *for all $M \in \mathcal{M}$, there exists $f \in \mathcal{F}$ such that $\|f - f_M^\star\|_{\infty, d_{\xi(M)}} = 0$ (or $\|f - f_M^\star\|_{\infty, \mu_{\xi(M)}} = 0$ if $f$ is defined on $\mathcal{S}$).*

*Algorithms from Theorem 1 that also take $\mathcal{F}$ as input must have $\max_{M \in \mathcal{M}} \left(J_M(\xi(M)) - \mathbb{E}\left[J_M(\widehat{\pi})\right]\right) \geq \frac{\gamma^2}{8}$.*

---

[9]Thus $f^\star$ is primarily dependent on $M \in \mathcal{M}$ since $\xi$ is also a function of $\mathcal{M}$.

## 5.2 LIMITATION OF THE LOWER BOUND

The major drawback of Theorem 1 is the use of "over-coverage"—our dataset covers states that are unreachable from the initial state. Another concern may be that while $\xi(M)$ can be optimal, the value function class $\mathcal{Q}$ does not contain the optimal value function $Q^\star$.However, the exclusion of $Q^\star$ is unavoidable since Xie & Jiang (2020a) has proposed an upper bound under the condition of $Q^\star$-realizability under weaker data assumptions compared with ours.

## 6 ON EXACTLY $Q^\star$-REALIZABILITY

The seminal work Xie & Jiang (2020a) presents an algorithm with state abstraction, showing that $Q^\star$-realizability, together with their refined data assumptions, is enough for learning near-optimal policies. Since the data assumption in Theorem 1 is stronger than the one from Xie & Jiang (2020a), Theorem 1 is, of course, incapable of $Q^\star$-realizability. This section bridges this gap by showing that even with $Q^\star$-realizability, learning near-optimal policies is impossible under the weaker partial coverage data assumption.

**Theorem 2.** *For any sample size $N$ and $\gamma \in [0, 1)$, there exist a family of MDPs $\mathcal{M}$ with the same state-action spaces, a reward function class $\mathcal{R}$, a transition kernel $\widetilde{P}$, a dataset $\mathcal{D}$ with distribution $d^\mathcal{D}$, a policy class $\Pi$, a state-action value function class $\mathcal{Q}$, a state transition probability class $\mathcal{P}_\mathcal{S}$, such that*

1. *$|\mathcal{A}| = |\mathcal{P}_\mathcal{S}| = |\Pi| = |\mathcal{R}| = 2, |\mathcal{Q}| = 8$,*

2. *for all $M \in \mathcal{M}$, $Q^\star_M \in \mathcal{Q}$ and $R_M \in \mathcal{R}$,*

3. *there exists a (unknown) mapping $\xi \in (\mathcal{M} \to \Pi)$ such that for all $M \in \mathcal{M}$,*

   - *$\big\|\frac{d_{\xi(M)}}{d^\mathcal{D}}\big\|_\infty \leq 32$,*

   - *$\xi(M)$ has its state-action value function realized by $\mathcal{Q}$ (i.e., $Q_{M,\xi(M)} \in \mathcal{Q}$),*

   - *the state transition kernel of $\xi(M)$ is realized by $\mathcal{P}_\mathcal{S}$ with exploratory-accuracy (i.e., $\exists P_\mathcal{S} \in \mathcal{P}_\mathcal{S}$, for all (possibly non-stationary) policy $\pi_{\text{non}}$, $\|P_\mathcal{S} - P_{M,\xi(M)}\|_{\infty,d_{\pi_{\text{non}}}} = 0$),*

   - *$\widetilde{P}$ is partially accurate under the induced distribution of $\xi(M)$ (i.e., $\|\widetilde{P} - P_M\|_{\infty,d_{\xi(M)}} = 0$).*

*Any learning algorithm $\mathfrak{A}$—which takes $\mathcal{S}, \mathcal{A}, \mathcal{R}, \mu_0, \gamma, \mathcal{D}, d^\mathcal{D}, \Pi, \mathcal{Q}, \widetilde{P}$ and $\mathcal{P}_\mathcal{S}$ as input and outputs a policy $\widehat{\pi}$—must satisfy*

$$\max_{M \in \mathcal{M}} \left( J_M(\xi(M)) - \mathbb{E}\big[J_M(\widehat{\pi})\big] \right) \geq \frac{\gamma^2}{16}.$$

The proof of Theorem 2 is provided in Appendix E.

A limitation of Theorem 2 is that while this counterexample considers $Q^\star$-realizability, the dataset does not cover the optimal policy. However, we argue that this result is still interesting since it allows the function class to contain the corresponding functions of the covered policy, and the dataset with partial coverage that does not cover optimal policies is more prevalent in real-world scenarios. Moreover, this theorem demonstrates the impossibility of achieving robust policy improvement with $Q^\star$-realizability.

## 7 CONCLUSION

This paper serves as an elaboration on offline RL with general function approximation. We clarify different types of function assumptions of offline RL—namely, the completeness-type and the realizability-type—by providing clear definitions and analyzing their practical usage. We interpret completeness-type assumptions as a typical requirement for the good approximation of targets under all possible policies considered in the algorithm. To determine the necessity of this requirement, we delve into the role of general function approximation in offline RL on learnability, and also propose a principle for establishing lower bounds. With the help of this principle, we propose two lower bounds showing the necessity of completeness-type assumptions.

## ACKNOWLEDGEMENTS

This work is supported by the National Science Fund for Distinguished Young Scholars (No. 62025602), the National Natural Science Foundation of China (No. U22B2036), the Tencent Foundation and XPLORER PRIZE, and Shanghai Artificial Intelligence Laboratory. We also thank anonymous reviewers for their valuable comments and suggestions.

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

## A  NOTATIONS

Table 1: Notations.

| | |
|---|---|
| $\mathcal{T}$ | Bellman operator, $\mathcal{T}f(s,a) := R(s,a) + \gamma\mathbb{E}_{s'\sim P(\cdot\mid s,a)}\big[\max f(s',\cdot)\big]$ |
| $\mathcal{T}^{\pi}$ | Bellman operator of policy $\pi$, $\mathcal{T}^{\pi}f(s,a) := R(s,a) + \gamma\mathbb{E}_{s'\sim P(\cdot\mid s,a),a'\sim\pi(\cdot\mid s')}\big[f(s',a')\big]$ |
| $\text{Unif}(X)$ | uniform distributions on $X$ (e.g., normalized Lebesgue measure for subsets of $\mathbb{R}^N$) |
| $\Delta(X)$ | distributions on $X$ |
| $\mathcal{S}$ | state space |
| $\mathcal{A}$ | action space |
| $\mathcal{Q}$ | state-action value function class |
| $\mathcal{W}$ | state-action density ratio function class |
| $V_{\pi}$ | state value function for policy $\pi$ |
| $Q_{\pi}$ | state-action value function for policy $\pi$ |
| $V^{\star}$ | optimal state value function |
| $Q^{\star}$ | optimal state-action value function |
| $d_{\pi}$ | induced distribution of a policy $\pi$, $d_{\pi}(s,a) := (1-\gamma)\big[\sum_{i=0}^{\infty}\gamma^i\mathbb{P}(s_i=s,a_i=a\mid s_0\sim\mu_0,\pi)\big]$ |
| $w_{\pi}$ | density ratio of $\pi$, $w_{\pi}(s,a) := d_{\pi}(s,a)/d^{\mathcal{D}}(s,a)$ |
| $\nu$ | uniform measure of $\mathcal{A}$, $\mathcal{S}$, or $\mathcal{S}\times\mathcal{A}$, depending on the context |
| $\mathcal{D}$ | dataset used in the algorithm |
| $d^{\mathcal{D}}$ | state-action distribution of the dataset |
| $\mu^{\mathcal{D}}$ | state distribution of dataset (margin of $d^{\mathcal{D}}$) |
| $\pi_q$ | policy that takes actions that maximize value function $q$ in each state, $\pi_q(s) := \text{argmax}_{a\in\mathcal{A}} q(s,\cdot)$ |
| $\pi_b$ | behaviour policy $\pi_b(s,a) := \begin{cases} d^{\mathcal{D}}(s,a)/\mu(s) & \text{if } \mu(s) > 0 \\ \nu(a) & \text{otherwise.} \end{cases}$ |
| $\mu_0$ | initial state distribution |
| $d_1 \gg d_2$ | $d_2$ is absolutely continuous w.r.t. $d_1$ |
| $\overline{E}$ | the complementary event of $E$ |
| $P_M^N$ | the law of events under MDP $M$ with size $N$, mainly used for datasets |
| $P_M$ | the law of events under MDP $M$ with size 1, mainly used for datasets |
| $P_{\pi}$ | $P_{\pi}(s'\mid s) := \int_{\mathcal{A}} P(s'\mid s,a)\pi(a\mid s)d\nu(a)$ |
| $\text{TV}(\mathbb{P},\mathbb{Q})$ | total variation distance between $\mathbb{P}$ and $\mathbb{Q}$ |
| $\chi^2(\mathbb{P},\mathbb{Q})$ | $\chi^2$-divergence between $\mathbb{P}$ and $\mathbb{Q}$ |
| $\|x\|_{p,q}$ | $q$-weighted $L^p$ norm, $\|x\|_{p,q} := \sqrt[p]{\int x^p dq}$ |

While $\pi, \mu$, and $d$ are mainly used to denote the Radon–Nikodym derivatives of the underlying probability measures w.r.t. $\nu$, we sometimes also use them to represent the corresponding distribution measures with a slight abuse of notations.

## B  HELPER LEMMAS

**Lemma 1** (Hypergeometric tail bound (Hoeffding, 1963; Skala, 2013)). *Let $X \sim\text{Hyper}(K, N, N')$[10] be a hypergeometric random variable, and define $p := K/N$. For any $pN' \geq \epsilon \geq 0$,*

$$\mathbb{P}[X - pN' \geq \epsilon N'] \leq \exp(-2\epsilon^2 N').$$

**Lemma 2.** *For any probability space defined with a sample space $\Omega$, a sigma-algebra $\mathcal{E}$, and probability measures $\mathbb{P}$ and $\mathbb{Q}$, the total variation distance between the conditional measures is upper bounded by the origin one:*

$$\forall\, E \in \mathcal{E},\ \text{TV}(\mathbb{P},\mathbb{Q}) \geq \text{TV}\big(\mathbb{P}(\cdot\mid E),\mathbb{Q}(\cdot\mid E)\big)\cdot\min\{\mathbb{P}(E),\mathbb{Q}(E)\}.$$

---

[10]We follow the definition from Wikipedia (2023): *the hypergeometric distribution is a discrete probability distribution that describes the probability of $X$ successes (random draws for which the object drawn has a specified feature) in $N'$ draws, without replacement, from a finite population of size $N$ that contains exactly $K$ objects with that feature, wherein each draw is either a success or a failure.*

*Proof.* For a fixed $E \in \mathcal{E}$, we define

$$\mathcal{E}' := \{E \cap E' \mid E' \in \mathcal{E}\}.$$

Since a sigma-algebra is closed under intersection, we have $\mathcal{E}' \subseteq \mathcal{E}$. Through the definition of total variation distance,

$$
\begin{aligned}
\mathrm{TV}(\mathbb{P}, \mathbb{Q}) &= \max_{E' \in \mathcal{E}} |\mathbb{P}(E') - \mathbb{Q}(E')| \\
&\geq \max_{E' \in \mathcal{E}'} |\mathbb{P}(E') - \mathbb{Q}(E')| \\
&= \max_{E' \in \mathcal{E}} |\mathbb{P}(E' \cap E) - \mathbb{Q}(E' \cap E)| \\
&\geq \max_{E' \in \mathcal{E}} |\mathbb{P}(E' \mid E)\mathbb{P}(E) - \mathbb{Q}(E' \mid E)\mathbb{Q}(E)| \\
&\geq \max_{E' \in \mathcal{E}} |\mathbb{P}(E' \mid E) - \mathbb{Q}(E' \mid E)| \cdot \min\{\mathbb{P}(E), \mathbb{Q}(E)\} \\
&= \mathrm{TV}(\mathbb{P}(\cdot \mid E), \mathbb{Q}(\cdot \mid E)) \cdot \min\{\mathbb{P}(E), \mathbb{Q}(E)\}.
\end{aligned}
$$

This completes the proof. $\qquad\square$

## C   ON EXACT CONSTRUCTION OF $\mathcal{F}(\mathcal{M}, \mathbb{A})$

In most cases, a function we want to approximate is a *property* summarized from a MDP. We can define a property (the concept of a function) as a mapping from a MDP class $\mathcal{M}$. Furthermore, suppose we have a function class $\mathcal{F}$ used to approximate properties in advance, an $\mathcal{F}$-property is a mapping from $\mathcal{M} \times \mathcal{F}$.[11] For example, the action space and the optimal value function are properties, and the value functions of policies from $\Pi$ are $\Pi$-properties. We can instantiate a property given one specific $M \in \mathcal{M}$ (or $f \in \mathcal{F}$ and $M \in \mathcal{M}$ for $\mathcal{F}$-property), such as computing the exact optimal value function of this $M$. The instantiation for the real model $M^\star$ is always the target we want to approximate.

The uniqueness of properties under any specific $M \in \mathcal{M}$ means that we can derive a function class for any properties by collecting their instantiations for each MDP from a realizable $\mathcal{M}$.

**Proposition 2.** *Given a model class $\mathcal{M}$ which contains the real model, for any property (denoted as $f$), we can construct a function class $\mathcal{F}$ with the realizability-type assumptions satisfied and $|\mathcal{F}| = |\mathcal{M}|$.*

**Proposition 3.** *Given a model class $\mathcal{M}$ which contains the real model and a function class $\mathcal{F}$, for any $\mathcal{F}$-property (denoted as $f'$), we can construct a function class $\mathcal{F}'$ with the completeness-type assumptions w.r.t. $\mathcal{F}$ satisfied and $|\mathcal{F}'| = |\mathcal{M}| \cdot |\mathcal{F}|$.*

Thus, among all assumptions, model-realizability ($M^\star \in \mathcal{M}$) is the strongest.

**Remark 6.** This tiered structure is incapable of completeness-type assumptions on one function class like Bellman-completeness. However, most completeness-type assumptions are built on different function classes, and even the Bellman-completeness can be reconstructed into a completeness-type assumption between two function classes (Chen & Jiang, 2019).

**Example 5** (Value functions of all policies). *The state-action value function is a $\Pi$-property. If there are a policy class $\Pi$ and a model class $\mathcal{M}$, we can calculate the value function of policy $\pi \in \Pi$ under MDP $M \in \mathcal{M}$ as $Q_{M,\pi}$ with integration. If we want a value function class containing all value functions for policies from $\Pi$, we can take $\{Q_{M,\pi} \mid M \in \mathcal{M}, \pi \in \Pi\}$.*

## D   PROOFS OF SECTION 5

This section provides a detailed proof for Section 5. Appendix D.1 and Appendix D.2 first introduce the MDPs and the function classes used in the counterexamples. Appendix D.3 then sketches the proof of Theorem 1 with the help of some high-level proofs. Appendix D.6 provides proofs for the corollaries in Section 5.1. The remainder of this section is then devoted to the proof of some lemmas.

---

[11]We assume that these mappings are known—we can compute the desired property if we have $M \in \mathcal{M}$ (and $h \in \mathcal{F}$).

Note that throughout the proof, we sometimes introduce the multiplication of decimal terms and integers. We assume implicitly that we tune the terms so that the result are integers if necessary.

## D.1   MDP STRUCTURES

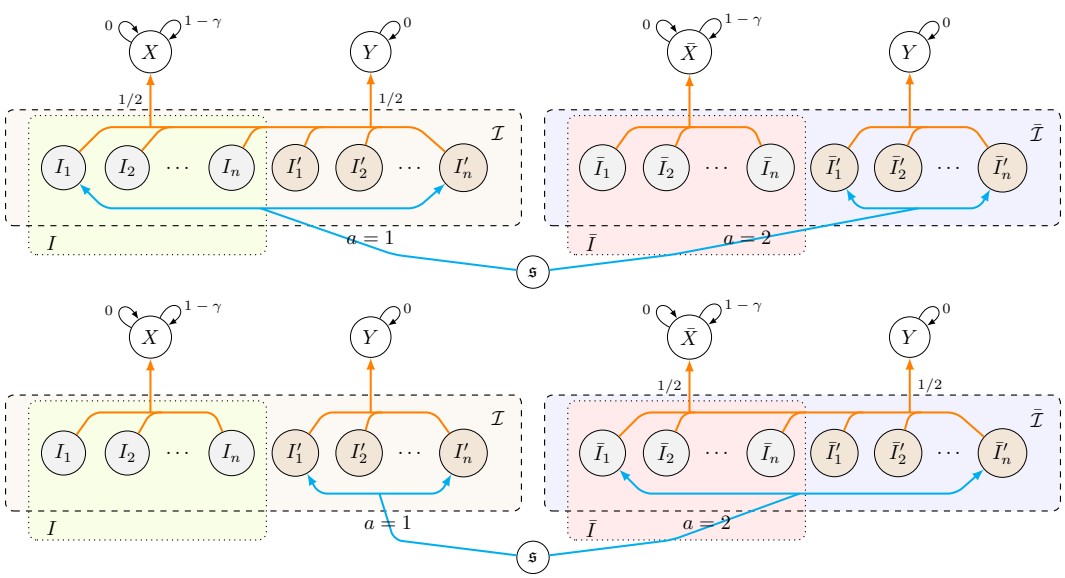

Figure 1: A counterexample from $\mathcal{M}_1$ (above) and a counterexample from $\mathcal{M}_2$ (below).

As shown in Figure 1, the constructed MDP class $\mathcal{M}$ contains $\mathcal{M}_1$ and its axisymmetric $\mathcal{M}_2$. MDPs in $\mathcal{M}_1$ and $\mathcal{M}_2$ only differ in their transition kernels.

**State space and action space**   As shown in Figure 1, $\mathcal{S}$ consists of an initial state $\mathfrak{s}$, a collection of intermediate states that belong to either set $\mathcal{I}$ or set $\overline{\mathcal{I}}$, and a set of terminal states $\{X, Y, \overline{X}\}$. The sets $\mathcal{I}$ and $\overline{\mathcal{I}}$ are disjoint, and have the same cardinality that will be specified later. We denote states from $\mathcal{I}$ as $\{I_1, I_2, I_3 \dots\}$, and states from $\overline{\mathcal{I}}$ as $\{\overline{I}_1, \overline{I}_2, \overline{I}_3 \dots\}$. We take $I$ (resp. $\overline{I}$) as a subset of $\mathcal{I}$ (resp. $\overline{\mathcal{I}}$) with half cardinality that would be used to determine the identities of the MDPs from $\mathcal{M}$. $I$ and $\overline{I}$ can be summarized by sets $\mathfrak{I} := \{I \subseteq \mathcal{I} \mid |I| = |\mathcal{I}|/2\}$ and $\overline{\mathfrak{I}} := \{\overline{I} \subseteq \overline{\mathcal{I}} \mid |\overline{I}| = |\overline{\mathcal{I}}|/2\}$. There are two actions ($\mathcal{A} = \{1, 2\}$) in each state.

**Transition kernel**   For MDPs from $\mathcal{M}_1$ with $\overline{I}$ specified,

$$P_1(\cdot \mid s, a) = \begin{cases} \text{Unif}(\mathcal{I}) & \text{if } s = \mathfrak{s} \text{ and } a = 1 \\ \text{Unif}(\overline{\mathcal{I}} \setminus \overline{I}) & \text{if } s = \mathfrak{s} \text{ and } a = 2 \\ \text{Unif}(\{X, Y\}) & \text{if } s \in \mathcal{I} \\ \text{Unif}(\{\overline{X}\}) & \text{if } s \in \overline{I} \\ \text{Unif}(\{Y\}) & \text{if } s \in \overline{\mathcal{I}} \setminus \overline{I} \\ \text{Unif}(\{s\}) & \text{otherwise.} \end{cases}$$

Also, for MDPs from $\mathcal{M}_2$, we "swap" the transition probability vertically,

$$P_2(\cdot \mid s, a) = \begin{cases} \text{Unif}(\mathcal{I} \setminus I) & \text{if } s = \mathfrak{s} \text{ and } a = 1 \\ \text{Unif}(\overline{\mathcal{I}}) & \text{if } s = \mathfrak{s} \text{ and } a = 2 \\ \text{Unif}(\{\overline{X}, Y\}) & \text{if } s \in \overline{\mathcal{I}} \\ \text{Unif}(\{X\}) & \text{if } s \in I \\ \text{Unif}(\{Y\}) & \text{if } s \in \mathcal{I} \setminus I \\ \text{Unif}(\{s\}) & \text{otherwise.} \end{cases}$$

As each $\bar{I} \in \bar{\mathfrak{I}}$ and each $I \in \mathfrak{I}$ will determinate the transition kernel for MDPs from $\mathcal{M}_1$ and $\mathcal{M}_2$, which is the only difference between them, $\mathcal{M}_1$ and $\mathcal{M}_2$ can actually be written as

$$\mathcal{M}_1 = \{M_{\bar{I}} \mid \bar{I} \in \bar{\mathfrak{I}}\} \quad \text{and} \quad \mathcal{M}_2 = \{M_I \mid I \in \mathfrak{I}\}.$$

**Reward**  One can only receive non-zero rewards by taking action 2 at states $\{X, \overline{X}\}$, which gives a $1 - \gamma$ reward:

$$R(s, a) = \begin{cases} 1 - \gamma & \text{if } s \in \{X, \overline{X}\} \text{ and } a = 2 \\ 0 & \text{otherwise.} \end{cases} \tag{3}$$

### D.2 Funciton classes and data distribution

**Policy function class**  The policy function class $\Pi$ contains two policies $\pi_1$ and $\pi_2$, which are given by

$$\pi_1(s) = \begin{cases} 2 & \text{if } s = X \\ 1 & \text{otherwise,} \end{cases} \quad \pi_2(s) = \begin{cases} 1 & \text{if } s = X \\ 2 & \text{otherwise.} \end{cases}$$

It is worth pointing out that, in the context of Theorem 1, $\pi_1$ is optimal for all the MDPs belonging to $\mathcal{M}_1$, and $\pi_2$ is optimal for all the MDPs from $\mathcal{M}_2$. $\xi$ thus maps MDPs in $\mathcal{M}_1$ to $\pi_1$, and maps MDPs in $\mathcal{M}_2$ to $\pi_2$ in this circumstance.

**Value function class**  The state-action value function class $\mathcal{Q}$ contains two functions $q_1$ and $q_2$, which are given by

$$q_1(s, a) = \begin{cases} \gamma^2/2 & \text{if } s = \mathfrak{s} \text{ and } a = 1 \\ 0 & \text{if } s = \mathfrak{s} \text{ and } a = 2 \\ \gamma/2 & \text{if } s \in \mathcal{I} \\ 0 & \text{if } s \in \bar{\mathcal{I}} \\ \gamma & \text{if } s = X \text{ and } a = 1 \\ 1 & \text{if } s = X \text{ and } a = 2 \\ 0 & \text{if } s = \overline{X} \text{ and } a = 1 \\ 1 - \gamma & \text{if } s = \overline{X} \text{ and } a = 2 \\ 0 & \text{if } s = Y, \end{cases} \quad q_2(s, a) = \begin{cases} 0 & \text{if } s = \mathfrak{s} \text{ and } a = 1 \\ \gamma^2/2 & \text{if } s = \mathfrak{s} \text{ and } a = 2 \\ 0 & \text{if } s \in \mathcal{I} \\ \gamma/2 & \text{if } s \in \bar{\mathcal{I}} \\ 0 & \text{if } s = X \text{ and } a = 1 \\ 1 - \gamma & \text{if } s = X \text{ and } a = 2 \\ \gamma & \text{if } s = \overline{X} \text{ and } a = 1 \\ 1 & \text{if } s = \overline{X} \text{ and } a = 2 \\ 0 & \text{if } s = Y. \end{cases}$$

One can check that $q_1$ is the value function of $\pi_1$ for MDPs in $\mathcal{M}_1$, and $q_2$ is the value function of $\pi_2$ for MDPs in $\mathcal{M}_2$.

**Partially accurate transition kernel**  We set $\widetilde{P}$ as

$$\widetilde{P}(\cdot \mid s, a) = \begin{cases} \text{Unif}(\mathcal{I}) & \text{if } s = \mathfrak{s} \text{ and } a = 1 \\ \text{Unif}(\bar{\mathcal{I}}) & \text{if } s = \mathfrak{s} \text{ and } a = 2 \\ \text{Unif}(\{X, Y\}) & \text{if } s \in \mathcal{I} \\ \text{Unif}(\{\overline{X}, Y\}) & \text{if } s \in \bar{\mathcal{I}} \\ \text{Unif}(\{s\}) & \text{otherwise.} \end{cases}$$

**State transition kernel with exploratory-accuracy**  The set $\mathcal{P}_\mathcal{S}$ contains two elements described below:

$$P_{\mathcal{S},1}(\cdot \mid s) = \begin{cases} \text{Unif}(\mathcal{I}) & \text{if } s = \mathfrak{s} \\ \text{Unif}(\{X, Y\}) & \text{if } s \in \mathcal{I} \\ \text{Unif}(\{Y\}) & \text{if } s \in \bar{\mathcal{I}} \\ \text{Unif}(\{s\}) & \text{otherwise,} \end{cases} \quad \text{and} \quad P_{\mathcal{S},2}(\cdot \mid s) = \begin{cases} \text{Unif}(\bar{\mathcal{I}}) & \text{if } s = \mathfrak{s} \\ \text{Unif}(\{\overline{X}, Y\}) & \text{if } s \in \bar{\mathcal{I}} \\ \text{Unif}(\{Y\}) & \text{if } s \in \mathcal{I} \\ \text{Unif}(\{s\}) & \text{otherwise.} \end{cases}$$

We point out that $P_{\mathcal{S},1}$ is the state transition kernel for $\pi_1$ under MDPs from $\mathcal{M}_1$, and $P_{\mathcal{S},2}$ is the transition kernel for $\pi_2$ under MDPs from $\mathcal{M}_2$.

**Remark 7.**  All functions mentioned above are independent with the choice of $I$ and $\bar{I}$.

**Data distribution** The state distribution is a uniform distribution on the "endpoint" and intermediate states repectively. The behaviour policy is a uniform distribution w.r.t. the action space $\mathcal{A}$, regardless of the underlying state,

$$d^{\mathcal{D}} = \frac{1}{2}\text{Unif}\Big(\{X, Y, \overline{X}, \mathfrak{s}\} \times \mathcal{A}\Big) + \frac{1}{2}\text{Unif}\Big((\mathcal{I} \cup \overline{\mathcal{I}}) \times \mathcal{A}\Big).$$

### D.3 PROOF SKETCH OF THEOREM 1

**From learning to hypothesis testing** Following the similar principle of deriving minimax lower bounds (Yu, 1997; Yang & Barron, 1999; Tsybakov, 2008), we begin with transferring an estimation problem to a hypothesis testing problem, of which the corresponding error probability can be related to the total variation distance between certain distributions.

First, the sub-optimality of the learned policies mainly comes from the chance of selecting action 2 in MDPs from $\mathcal{M}_1$, and the chance of selecting action 1 in MDPs from $\mathcal{M}_2$. For MDP $M \in \mathcal{M}$ and learning algorithm $\mathfrak{A}$, we denote this event as $E_M^{\mathfrak{A}}$. Letting $\widetilde{\pi}_M = \pi_1$ for $M \in \mathcal{M}_1$ and $\widetilde{\pi}_M = \pi_2$ for $M \in \mathcal{M}_2$. For any $M \in \mathcal{M}$, we have

$$J_M(\widetilde{\pi}_M) - \mathbb{E}[J_M(\widehat{\pi})] \geq \gamma^2/2 - \frac{\gamma^2}{2}\mathbb{P}_M^N(\overline{E}_M^{\mathfrak{A}}) = \frac{\gamma^2}{2}\mathbb{P}_M^N(E_M^{\mathfrak{A}}). \tag{4}$$

Furthermore, we transfer the minimax lower bound into a Bayesian problem,

$$\max_{M \in \mathcal{M}} \big[\mathbb{P}_M^N(E_M^{\mathfrak{A}})\big] \geq \mathbb{E}_{M \sim \text{Unif}(\mathcal{M})}\big[\mathbb{P}_M^N(E_M^{\mathfrak{A}})\big]. \tag{5}$$

For any learning algorithm $\mathfrak{A}$, we have

$$\mathbb{E}_{M \sim \text{Unif}(\mathcal{M})}\big[\mathbb{P}_M^N(E_M^{\mathfrak{A}})\big] = \frac{1}{2}\mathbb{E}_{M \sim \text{Unif}(\mathcal{M}_1)}\big[\mathbb{P}_M^N(E_M^{\mathfrak{A}})\big] + \frac{1}{2}\mathbb{E}_{M \sim \text{Unif}(\mathcal{M}_2)}\big[\mathbb{P}_M^N(E_M^{\mathfrak{A}})\big] \tag{6}$$

$$\geq \frac{1}{2}\left(1 - \text{TV}\left(\frac{1}{|\mathcal{M}_1|}\sum_{M \in \mathcal{M}_1}\mathbb{P}_M^N, \frac{1}{|\mathcal{M}_2|}\sum_{M \in \mathcal{M}_2}\mathbb{P}_M^N\right)\right), \tag{7}$$

where Eqn. (7) follows from (i) the inequality $\mathbb{P}(E) + \mathbb{Q}(\overline{E}) \geq 1 - \text{TV}(\mathbb{P}, \mathbb{Q})$, and (ii) the event $E_M^{\mathfrak{A}}$ when $M \in \mathcal{M}_1$ is the complement with itself when $M \in \mathcal{M}_2$. While the total variation distance from Eqn. (7) is algorithm dependent, we can extend it as a probability on the dataset $\mathcal{D}$ through data processing inequality, for which we add subscript $\mathcal{D}$ for clarify:

$$\text{TV}\left(\frac{1}{|\mathcal{M}_1|}\sum_{M \in \mathcal{M}_1}\mathbb{P}_M^N, \frac{1}{|\mathcal{M}_2|}\sum_{M \in \mathcal{M}_2}\mathbb{P}_M^N\right) \leq \text{TV}_{\mathcal{D}}\left(\frac{1}{|\mathcal{M}_1|}\sum_{M \in \mathcal{M}_1}\mathbb{P}_M^N, \frac{1}{|\mathcal{M}_2|}\sum_{M \in \mathcal{M}_2}\mathbb{P}_M^N\right). \tag{8}$$

Combining Eqns. (4), (5), (7) and (8), we obtain

$$\inf_{\mathfrak{A}} \sup_{M \in \mathcal{M}} \left[J_M(\widetilde{\pi}_M) - \mathbb{E}_{\mathcal{D}_N \sim \mathbb{P}_{M,N}}[J_M(\widehat{\pi})]\right] \geq \frac{\gamma^2}{4}\left(1 - \text{TV}_{\mathcal{D}}\left(\frac{1}{|\mathcal{M}_1|}\sum_{M \in \mathcal{M}_1}\mathbb{P}_M^N, \frac{1}{|\mathcal{M}_2|}\sum_{M \in \mathcal{M}_2}\mathbb{P}_M^N\right)\right). \tag{9}$$

**Bounding the Total variation distance** For notational convenience, we define

$$\mathbb{P}_1^N := \frac{1}{|\mathcal{M}_1|}\sum_{M \in \mathcal{M}_1}\mathbb{P}_M^N \quad \text{and} \quad \mathbb{P}_2^N := \frac{1}{|\mathcal{M}_2|}\sum_{M \in \mathcal{M}_2}\mathbb{P}_M^N. \tag{10}$$

The remaining task is to bound the total variation distance between $\mathbb{P}_1^N$ and $\mathbb{P}_2^N$, and it is done through the following lemma.

**Lemma 3.** *For any sample size $N$, we can choose the cardinality of $\mathcal{I}$ to achieve*

$$\text{TV}_{\mathcal{D}}(\mathbb{P}_1^N, \mathbb{P}_2^N) \leq 1/2. \tag{11}$$

The proof of Lemma 3 is provided in Appendix D.5.

Finally, combining Eqns. (9) and (11) yields that

$$\inf_{\mathfrak{A}} \sup_{M \in \mathcal{M}} \left[J_M(\widetilde{\pi}_M) - \mathbb{E}[J_M(\widehat{\pi})]\right] \geq \frac{\gamma^2}{8}.$$

This completes the proof.

### D.4 A GENERAL BOUND

**Lemma 4.** *For any* $0 \leq \epsilon \leq |\overline{\mathcal{I}}|/4$,

$$\mathrm{TV}_{\mathcal{D}}\left(\mathbb{P}_1^N, \mathbb{P}_2^N\right) \leq \sqrt{\left(1 + \frac{5\epsilon}{8}\right)^N + \exp\left(-\epsilon^2|\overline{\mathcal{I}}| + N\log\left(\frac{21}{16}\right)\right) - 1}.$$

The proof of Lemma 4 is adapted from Foster et al. (2021b) with appropriate modifications. Specifically, we first construct a *dominating measure* that follows the left branch of $\mathcal{M}_1$ and the right branch of $\mathcal{M}_2$, then we transfer the total variance distance to $\chi^2$-divergence and bound it via the hypergeometric tail bound.

*Proof.*

**Construct the dominating measure** Because of the use of $\chi^2$-divergence, we first construct an MDP $M_0$ whose transition kernel is

$$P_0(\cdot \mid s, a) = \begin{cases} \mathrm{Unif}(\mathcal{I}) & \text{if } s = \mathfrak{s} \text{ and } a = 1 \\ \mathrm{Unif}(\overline{\mathcal{I}}) & \text{if } s = \mathfrak{s} \text{ and } a = 2 \\ \mathrm{Unif}(\{X, Y\}) & \text{if } s \in \mathcal{I} \\ \mathrm{Unif}(\{\overline{X}, Y\}) & \text{if } s \in \overline{\mathcal{I}} \\ \mathrm{Unif}(\{s\}) & \text{otherwise} \end{cases}$$

and whose reward function follows Eqn. (3) (as shown in Figure 2). We define the data collecting process under $M_0$ and the data distribution $d^{\mathcal{D}}$ with sample size $N$ as $\mathbb{P}_0^N$.

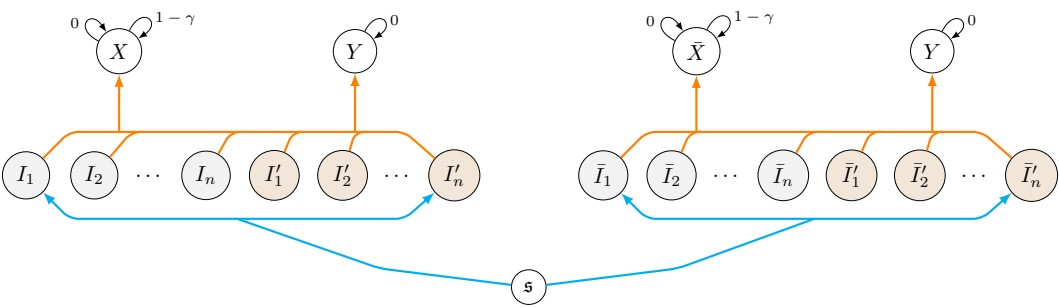

Figure 2: The MDP of the dominating measure.

**From total variance distance to $\chi^2$-divergence** We first decouple the total variance distance via the triangular inequality and transfer it to $\chi^2$-divergence,[12]

$$\mathrm{TV}_{\mathcal{D}}\left(\mathbb{P}_1^N, \mathbb{P}_2^N\right) \leq \mathrm{TV}_{\mathcal{D}}\left(\mathbb{P}_1^N, \mathbb{P}_0^N\right) + \mathrm{TV}_{\mathcal{D}}\left(\mathbb{P}_2^N, \mathbb{P}_0^N\right)$$

$$\leq \frac{1}{2}\sqrt{\chi^2\left(\mathbb{P}_1^N, \mathbb{P}_0^N\right)} + \frac{1}{2}\sqrt{\chi^2\left(\mathbb{P}_2^N, \mathbb{P}_0^N\right)}.$$

---

[12]We refer the interested readers to Polyanskiy & Wu (2022) for more details about these inequalities.

As the last two terms in the above inequality are symmetric, we will introduce how to bound the first term, while skipping the proof for the second term for simplicity. By the definition of $\chi^2$-divergence,

$$
\chi^2\left(\mathbb{P}_1^N, \mathbb{P}_0^N\right) = \mathbb{E}_{\mathcal{D}\sim\mathbb{P}_0^N}\left[\frac{\left(\frac{1}{|\mathcal{M}_1|}\sum_{M\in\mathcal{M}_1}\mathbb{P}_M^N(\mathcal{D})\right)^2}{\left(\mathbb{P}_0^N(\mathcal{D})\right)^2}\right] - 1
$$

$$
= \mathbb{E}_{\mathcal{D}\sim\mathbb{P}_0^N}\left[\frac{\frac{1}{|\mathcal{M}_1|^2}\sum_{M,M'\in\mathcal{M}_1}\mathbb{P}_M^N(\mathcal{D})\mathbb{P}_{M'}^N(\mathcal{D})}{\left(\mathbb{P}_0^N(\mathcal{D})\right)^2}\right] - 1
$$

$$
= \frac{1}{|\mathcal{M}_1|^2}\sum_{M,M'\in\mathcal{M}_1}\mathbb{E}_{\mathcal{D}\sim\mathbb{P}_0^N}\left[\frac{\mathbb{P}_M^N(\mathcal{D})\mathbb{P}_{M'}^N(\mathcal{D})}{\left(\mathbb{P}_0^N(\mathcal{D})\right)^2}\right] - 1
$$

$$
= \frac{1}{|\mathcal{M}_1|^2}\sum_{M,M'\in\mathcal{M}_1}\mathbb{E}_{\{(s_i,a_i,r_i,s_i')\}_{i=1}^N\sim\mathbb{P}_0^N}\left[\frac{\prod_{i=1}^N\left(\mathbb{P}_M(s_i,a_i,r_i,s_i')\mathbb{P}_{M'}(s_i,a_i,r_i,s_i')\right)}{\prod_{i=1}^N\left(\mathbb{P}_0(s_i,a_i,r_i,s_i')\right)^2}\right] - 1
$$

$$
= \frac{1}{|\mathcal{M}_1|^2}\sum_{M,M'\in\mathcal{M}_1}\mathbb{E}_{\{(s_i,a_i,r_i,s_i')\}_{i=1}^N\sim\mathbb{P}_0^N}\left[\prod_{i=1}^N\frac{\left(\mathbb{P}_M(s_i,a_i,r_i,s_i')\mathbb{P}_{M'}(s_i,a_i,r_i,s_i')\right)}{\left(\mathbb{P}_0(s_i,a_i,r_i,s_i')\right)^2}\right] - 1.
$$

Since the dataset is generated in an i.i.d. manner, we can take the multiplication out from the expectation,

$$
\chi^2\left(\mathbb{P}_1^N, \mathbb{P}_0^N\right) = \frac{1}{|\mathcal{M}_1|^2}\sum_{M,M'\in\mathcal{M}_1}\left(\mathbb{E}_{(s,a,r,s')\sim\mathbb{P}_0}\left[\frac{\mathbb{P}_M(s,a,r,s')\mathbb{P}_{M'}(s,a,r,s')}{\left(\mathbb{P}_0(s,a,r,s')\right)^2}\right]\right)^N - 1, \quad (12)
$$

where $\mathbb{P}_0$ and $\mathbb{P}_1$ are the one sample version of $\mathbb{P}_0^N$ and $\mathbb{P}_1^N$, correspondingly. Through the definition of the data collecting process, for any $M, M' \in \mathcal{M}_1$, we have

$$
\mathbb{E}_{(s,a,r,s')\sim\mathbb{P}_0}\left[\frac{\mathbb{P}_M(s,a,r,s')\mathbb{P}_{M'}(s,a,r,s')}{\left(\mathbb{P}_0(s,a,r,s')\right)^2}\right]
$$

$$
= \frac{11}{16} + \frac{1}{16}\mathbb{E}_{r=R(\mathfrak{s},2),s'\sim\mathbb{P}_0(\cdot|\mathfrak{s},2)}\left[\frac{\mathbb{P}_M(\mathfrak{s},2,r,s')\mathbb{P}_{M'}(\mathfrak{s},2,r,s')}{\left(\mathbb{P}_0(\mathfrak{s},2,r,s')\right)^2}\right]
$$

$$
+ \frac{1}{4}\mathbb{E}_{(s,a)\sim\text{Unif}(\bar{\mathcal{I}}\times\mathcal{A}),r=R(s,a),s'\sim\mathbb{P}_0(\cdot|s,a)}\left[\frac{\mathbb{P}_M(s,a,r,s')\mathbb{P}_{M'}(s,a,r,s')}{\left(\mathbb{P}_0(s,a,r,s')\right)^2}\right]
$$

$$
= \frac{11}{16} + \frac{1}{16}\underbrace{\mathbb{E}_{s'\sim\mathbb{P}_0(\cdot|\mathfrak{s},2)}\left[\frac{\mathbb{P}_M(s'\mid\mathfrak{s},2)\mathbb{P}_{M'}(s'\mid\mathfrak{s},2)}{\left(\mathbb{P}_0(s'\mid\mathfrak{s},2)\right)^2}\right]}_{\Phi_1(t_{M,M'})}
$$

$$
+ \frac{1}{4}\underbrace{\mathbb{E}_{(s,a)\sim\text{Unif}(\bar{\mathcal{I}}\times\mathcal{A}),s'\sim\mathbb{P}_0(\cdot|s,a)}\left[\frac{\mathbb{P}_M(s'\mid s,a)\mathbb{P}_{M'}(s'\mid s,a)}{\left(\mathbb{P}_0(s'\mid s,a)\right)^2}\right]}_{\Phi_2(t_{M,M'})}.
$$

The last equality holds since the reward function is ths same. As the last two variables only depend on the number of overlapped states between $\bar{I}_M$ and $\bar{I}_{M'}$, which we denote as $t_{M,M'}$, we can define them as $\Phi_1(t_{M,M'})$ and $\Phi_2(t_{M,M'})$ respectively.

**Model as hyper-geometric distribution** We take the summation in Eqn. (12) as taking the expectation of sampling two MDPs uniformly from $\mathcal{M}_1$,

$$\frac{1}{|\mathcal{M}_1|^2} \sum_{M,M' \in \mathcal{M}_1} \left( \mathbb{E}_{(s,a,r,s') \sim \mathbb{P}_0} \left[ \frac{\mathbb{P}_M(s,a,r,s')\mathbb{P}_{M'}(s,a,r,s')}{\left(\mathbb{P}_0(s,a,r,s')\right)^2} \right] \right)^N - 1$$

$$= \mathbb{E}_{M \sim \text{Unif}(\mathcal{M}_1)} \mathbb{E}_{M' \sim \text{Unif}(\mathcal{M}_1)} \left[ \left( \mathbb{E}_{(s,a,r,s') \sim \mathbb{P}_0} \left[ \frac{\mathbb{P}_M(s,a,r,s')\mathbb{P}_{M'}(s,a,r,s')}{\left(\mathbb{P}_0(s,a,r,s')\right)^2} \right] \right)^N \right] - 1$$

$$= \mathbb{E}_{M \sim \text{Unif}(\mathcal{M}_1)} \mathbb{E}_{M' \sim \text{Unif}(\mathcal{M}_1)} \left[ \left( \frac{11}{16} + \frac{1}{16}\Phi_1(t_{M,M'}) + \frac{1}{4}\Phi_2(t_{M,M'}) \right)^N \right] - 1.$$

As MDPs from $\mathcal{M}_1$ are indexed by $\bar{I} \in \bar{\mathfrak{I}}$, for any fixed $M \in \mathcal{M}_1$,

$$\mathbb{E}_{M' \sim \text{Unif}(\mathcal{M}_1)} \left[ \left( \frac{11}{16} + \frac{1}{16}\Phi_1(t_{M,M'}) + \frac{1}{4}\Phi_2(t_{M,M'}) \right)^N \right] - 1$$

is equal to taking the expectation for a function of $t_{M,M'}$ w.r.t. the following process:

*Fix $\bar{I}_M$, we are selecting $\bar{I}_{M'}$ without replacement uniformly from $\bar{\mathcal{I}}$. Take the number of overlapped state between $\bar{I}_M$ and $\bar{I}_{M'}$ as $t_{M,M'}$, and denote this random variable as $T$.*

Through the definition, we have $T \sim \text{Hyper}(0.5|\bar{\mathcal{I}}|, |\bar{\mathcal{I}}|, 0.5|\bar{\mathcal{I}}|)$ for any choice of $M$ and $M'$. Thus,

$$\mathbb{E}_{M \sim \text{Unif}(\mathcal{M}_1)} \mathbb{E}_{M' \sim \text{Unif}(\mathcal{M}_1)} \left[ \left( \frac{11}{16} + \frac{1}{16}\Phi_1(t_{M,M'}) + \frac{1}{4}\Phi_2(t_{M,M'}) \right)^N \right]$$

$$= \mathbb{E}_{M \sim \text{Unif}(\mathcal{M}_1)} \mathbb{E}_{T \sim \text{Hyper}(0.5|\bar{\mathcal{I}}|, |\bar{\mathcal{I}}|, 0.5|\bar{\mathcal{I}}|)} \left[ \left( \frac{11}{16} + \frac{1}{16}\Phi_1(T) + \frac{1}{4}\Phi_2(T) \right)^N \right]$$

$$= \mathbb{E}_{T \sim \text{Hyper}(0.5|\bar{\mathcal{I}}|, |\bar{\mathcal{I}}|, 0.5|\bar{\mathcal{I}}|)} \left[ \left( \frac{11}{16} + \frac{1}{16}\Phi_1(T) + \frac{1}{4}\Phi_2(T) \right)^N \right].$$

By definition,

$$\Phi_1(\theta|\bar{\mathcal{I}}|) = \frac{\theta}{0.5^2} = 4\theta, \quad \Phi_2(\theta|\bar{\mathcal{I}}|) = 0.5\frac{\theta}{0.5^2} + (1 - 0.5)\frac{\theta}{(1 - 0.5)^2} = 4\theta. \qquad (13)$$

Note that both of them are monotonic.

**Bound via exponential tail bound** Eqn. (12) is finite only if the probability that

$$\frac{11}{16} + \frac{1}{16}\Phi_1(T) + \frac{1}{4}\Phi_2(T) > 1$$

is exponentially small. We divide the expectation into two parts,

$$
\mathbb{E}_{T \sim \text{Hyper}(0.5|\bar{\mathcal{I}}|, |\bar{\mathcal{I}}|, 0.5|\bar{\mathcal{I}}|)} \left[ \left( \frac{11}{16} + \frac{1}{16}\Phi_1(T) + \frac{1}{4}\Phi_2(T) \right)^N \right]
$$

$$
= \sum_{t=0}^{0.5|\bar{\mathcal{I}}|} \frac{\binom{0.5|\bar{\mathcal{I}}|}{t}\binom{0.5|\bar{\mathcal{I}}|}{0.5|\bar{\mathcal{I}}|-t}}{\binom{|\bar{\mathcal{I}}|}{0.5|\bar{\mathcal{I}}|}} \left( \frac{11}{16} + \frac{1}{16}\Phi_1(t) + \frac{1}{4}\Phi_2(t) \right)^N
$$

$$
= \sum_{t=0}^{(0.5+\epsilon)0.5|\bar{\mathcal{I}}|} \frac{\binom{0.5|\bar{\mathcal{I}}|}{t}\binom{0.5|\bar{\mathcal{I}}|}{0.5|\bar{\mathcal{I}}|-t}}{\binom{|\bar{\mathcal{I}}|}{0.5|\bar{\mathcal{I}}|}} \left( \frac{11}{16} + \frac{1}{16}\Phi_1(t) + \frac{1}{4}\Phi_2(t) \right)^N
$$

$$
+ \sum_{t=(0.5+\epsilon)0.5|\bar{\mathcal{I}}|}^{0.5|\bar{\mathcal{I}}|} \frac{\binom{0.5|\bar{\mathcal{I}}|}{t}\binom{0.5|\bar{\mathcal{I}}|}{0.5|\bar{\mathcal{I}}|-t}}{\binom{|\bar{\mathcal{I}}|}{0.5|\bar{\mathcal{I}}|}} \left( \frac{11}{16} + \frac{1}{16}\Phi_1(t) + \frac{1}{4}\Phi_2(t) \right)^N
$$

$$
\leq \underbrace{\left( \frac{11}{16} + \frac{1}{16}\Phi_1((0.5+\epsilon)0.5|\bar{\mathcal{I}}|) + \frac{1}{4}\Phi_2((0.5+\epsilon)0.5|\bar{\mathcal{I}}|) \right)^N}_{(1)}
$$

$$
+ \underbrace{\mathbb{P}(T \geq (0.5+\epsilon)0.5|\bar{\mathcal{I}}|) \left( \frac{11}{16} + \frac{1}{16}\Phi_1(0.5|\bar{\mathcal{I}}|) + \frac{1}{4}\Phi_2(0.5|\bar{\mathcal{I}}|) \right)^N}_{(2)}
$$

By the definition of $\Phi_1$ and $\Phi_2$, we have

$$
\left( \frac{11}{16} + \frac{1}{16}\Phi_1((0.5+\epsilon)0.5|\bar{\mathcal{I}}|) + \frac{1}{4}\Phi_2((0.5+\epsilon)0.5|\bar{\mathcal{I}}|) \right)^N
$$

$$
= \left( \frac{11}{16} + \frac{(0.5+\epsilon)0.5}{16 \cdot 0.5^2} + 0.5\frac{(0.5+\epsilon)0.5}{4 \cdot 0.5^2} + (1-0.5)\frac{1-0.5-0.5+0.5(0.5+\epsilon)}{4(1-0.5)^2} \right)^N
$$

(by definition)

$$
= \left( \frac{11}{16} + \frac{1}{16} + \frac{\epsilon}{8} + \frac{1}{8} + \frac{\epsilon}{4} + \frac{0.25+0.5\epsilon}{2} \right)^N
$$

$$
= \left( 1 + \frac{5\epsilon}{8} \right)^N,
$$

and

$$
\left( \frac{11}{16} + \frac{1}{16}\Phi_1(0.5|\bar{\mathcal{I}}|) + \frac{1}{4}\Phi_2(0.5|\bar{\mathcal{I}}|) \right)^N
$$

$$
= \left( \frac{11}{16} + \frac{0.5}{16 \cdot 0.5^2} + 0.5\frac{0.5}{4 \cdot 0.5^2} + (1-0.5)\frac{1-0.5-0.5+0.5}{4(1-0.5)^2} \right)^N
$$

$$
= \left( \frac{11}{16} + \frac{1}{8} + \frac{1}{4} + \frac{1}{4} \right)^N
$$

$$
= \left( \frac{21}{16} \right)^N.
$$

The probability presented in term (2) is exponentially small due to the hypergeometric tail bound (Lemma 1), which yields that

$$\mathbb{P}(T \geq (0.5 + \epsilon)0.5|\bar{\mathcal{I}}|)\left( \frac{11}{16} + \frac{1}{16}\Phi_1(0.5|\bar{\mathcal{I}}|) + \frac{1}{4}\Phi_2(0.5|\bar{\mathcal{I}}|) \right)^N$$

$$\leq \exp(-2\epsilon^2 \cdot 0.5|\bar{\mathcal{I}}|)\left( \frac{21}{16} \right)^N \qquad\qquad (0 \leq \epsilon \leq \tfrac{1}{4}|\bar{\mathcal{I}}|)$$

$$\leq \exp\left( -\epsilon^2|\bar{\mathcal{I}}| + N\log\left( \frac{21}{16} \right) \right).$$

Combining them, we have

$$\chi^2\left( \mathbb{P}_1^N, \mathbb{P}_0^N \right) \leq \left( 1 + \frac{5\epsilon}{8} \right)^N + \exp\left( -\epsilon^2|\bar{\mathcal{I}}| + N\log\left( \frac{21}{16} \right) \right) - 1.$$

Through the similar arguments, we also have

$$\chi^2\left( \mathbb{P}_2^N, \mathbb{P}_0^N \right) \leq \left( 1 + \frac{5\epsilon}{8} \right)^N + \exp\left( -\epsilon^2|\mathcal{I}| + N\log\left( \frac{21}{16} \right) \right) - 1.$$

Thus by noting that $|\mathcal{I}| = |\bar{\mathcal{I}}|$, we have

$$\mathrm{TV}_{\mathcal{D}}\left( \mathbb{P}_1^N, \mathbb{P}_2^N \right) \leq \sqrt{ \left( 1 + \frac{5\epsilon}{8} \right)^N + \exp\left( -\epsilon^2|\mathcal{I}| + N\log\left( \frac{21}{16} \right) \right) - 1 }.$$

This completes the proof. □

### D.5 Proof of Lemma 3

**Lemma** (Restatement of Lemma 3). *For any sample size $N$, we can choose the cardinality of $\mathcal{I}$ to achieve*

$$\mathrm{TV}_{\mathcal{D}}(\mathbb{P}_1^N, \mathbb{P}_2^N) \leq 1/2.$$

This lemma is a direct consequence of Lemma 4.

*Proof.* From Lemma 4, we have

$$\mathrm{TV}_{\mathcal{D}}(\mathbb{P}_1^N, \mathbb{P}_2^N) \leq \sqrt{ \left( 1 + \frac{5\epsilon}{8} \right)^N + \exp\left( -\epsilon^2|\bar{\mathcal{I}}| + N\log\left( \frac{21}{16} \right) \right) - 1 }$$

$$\leq \sqrt{ \exp\left( \frac{5N\epsilon}{8} \right) + \exp\left( -\epsilon^2|\bar{\mathcal{I}}| + N\log\left( \frac{21}{16} \right) \right) - 1 }.$$

Taking $0 < \epsilon \leq \frac{8}{5N}\log(9/8)$ and $|\bar{\mathcal{I}}| = |\mathcal{I}| \geq \frac{\left( N\log\left( \frac{21}{16} \right) + \log 8 \right)}{\epsilon^2}$ yields that

$$\mathrm{TV}_{\mathcal{D}}(\mathbb{P}_1^N, \mathbb{P}_2^N) \leq 1/2.$$

This completes the proof. □

### D.6 Proof of the corollaries

We can prove Corollary 1 by constructing the density ratio class from the state transition kernel class.

*Proof of Corollary 1.* For each state transition kernel, there is a $\mathcal{S}$-sequence $\{s_0, s_1, s_2, \ldots, s_i, \ldots\}$ such $s_0 \sim \mu_0, s_{i+1} \sim P_{\mathcal{S}}(\cdot \mid s_i)$. We can compute the state distribution of a state transition kernel $P_{\mathcal{S}}$ as

$$\mu(s) = (1 - \gamma) \sum_{i=0}^{\infty} \gamma^i \mathbb{P}(s_i = s).$$

If $P_{\mathcal{S}}$ is the state transition kernel of $\pi$, $\mu$ is the induced state distribution of $\pi$ (i.e., $\mu_\pi$). Moreover, $\mu$ is consistent if $P_{\mathcal{S}}$ is only accurate under $\mu_\pi$, i.e., have the same value with the real one on the support of $\mu_\pi$. The induced distribution of $\pi$ would be $d_\pi(s, a) = \mu(s) \cdot \pi(s, a)$, and the density ratio $w_\pi$ follows that

$$w_\pi(s, a) = \frac{d_\pi(s, a)}{d^{\mathcal{D}}(s, a)} = \frac{\mu(s)\pi(s, a)}{d^{\mathcal{D}}(s, a)}.$$

We can take the above process as a mapping from state transition kernels and policies to density ratios, and denote it as $O$. We can construct the density ratio class as $\mathcal{W} = \{O(P_{\mathcal{S}}, \pi) \mid P_{\mathcal{S}} \in \mathcal{P}_{\mathcal{S}}, \pi \in \Pi\}$.

For any $M \in \mathcal{M}$, due to the existence of exploratory-accurate state transition kernel of $\xi(M)$ in $\mathcal{P}_{\mathcal{S}}$, $\mathcal{W}$ must contain the density ratio of $\xi(M)$. $\square$

Since the function in Corollaries 2 and 3 are derived from trajectories, the proofs for them are mainly devoted to construct a set of functions mapping from $\mathcal{S} \times \mathcal{A}$ (or $\mathcal{S}$) to the trajectories (or the $\mathcal{S}$-trajectories) thereby.[13] Specifically, we want a function class $\mathcal{H}$ such that

for all $M \in \mathcal{M}$, there exists $h \in \mathcal{H}$ that can return accurate trajectories of $\xi(M)$ in certain level.

This is true due to the fact that we can calculate trajectories directly from the transition kernel.

*Proof of Corollary 2.* Exploratory-accuracy requries that for any entry reachable from $\mu_0$ by a non-stationary policy, the value of a function is the same with the target. For any entry reachable from the inital distirbution distribution by a specific non-stationary policy $\pi_{\mathrm{non}}$, we can denote the state of this entry as $s_e \in \mathcal{S}$. Given a state transition kernel $P_{\mathcal{S}}$, there is a $\mathcal{S}$-sequence

$$s_e, s_{e+1}, s_{e+2}, \ldots, s_{e+i}, \ldots$$

such that $s_{e+i+1} \sim P_{\mathcal{S}}(\cdot|s_{e+i})$. The $\mathcal{S}$-trajecroty from this entry can be caculated with the $\mathcal{S}$-sequence and policy $\pi$ as

$$s_e, r_e, \ldots, s_{e+i}, r_{e+i} \cdots \tag{14}$$

such that $r_{e+i} = \mathbb{E}_{a \sim \pi(\cdot|s_{e+i})}\big[R(s_{e+i}, a)\big]$.

Given policy $\pi$, Eqn. (14) is the $\mathcal{S}$-trajectory from $s_e$ induced by $\pi$ if $P_{\mathcal{S}}$ has the same value with $P_\pi$ on the states of real $\mathcal{S}$-trajectory from $s_e$ induced by $\pi$. This can always be achieved if $P_{\mathcal{S}}$ is exploratory-accurate w.r.t. $P_\pi$.

We can take the above process as a mapping from state transition kernels and policies to a function from $\mathcal{S}$ to trajectories, and denote it as $O$.

We then construct a set as $\{O(P_{\mathcal{S}}, \pi) \mid P_{\mathcal{S}} \in \mathcal{P}_{\mathcal{S}}, \pi \in \Pi\}$. For any $M \in \mathcal{M}$, any $(\xi(M), \mathcal{S})$-related function can be derived from one element from this class with exploratory-accuracy. $\square$

*Proof of Corollary 3.* Behaviour-accuracy w.r.t. a policy $\pi$ requries that for any entry reachable from $\mu_0$ by $\pi$, the value of a function is the same with the target. For any entry reachable from the inital distirbution distribution by $\pi$, we can denote this entry as $s_e$ or $(s_e, a_e)$ depending on the context. Given a transition kernel $P$ and a policy $\pi$, there is a trajectory

$$s_e, a_e, r_e, s_{e+1}, a_{e+1}, r_{e+1}, \ldots, s_{e+i}, a_{e+i}, r_{e+i} \cdots$$

for the $s_e$ entry such that $a_{e+i} \sim \pi(\cdot|s_{e+i}), r_{e+i} = R(s_{e+i}, a_{e+i})$ and $s_{e+i+1} \sim P(\cdot|s_{e+i}, a_{e+i})$ for all $i \geq 0$. For the $(s_e, a_e)$ entry, there is also a trajectory

$$s_e, a_e, r_e, s_{e+1}, a_{e+1}, r_{e+1}, \ldots, s_{e+i}, a_{e+i}, r_{e+i} \cdots$$

---

[13]Probability of trajectories, to be more accurate.

such that $a_{e+i} \sim \pi(\cdot|s_{e+i}), r_{e+i} = R(s_{e+i}, a_{e+i})$, and $s_{e+i+1} \sim P(\cdot|s_{e+i}, a_{e+i})$ for all $i \geq 0$, except that we fix the value of $a_e$.

The trajectory above is the trajectory from $s_e$ (or $(s_e, a_e)$ respectively) induced by $\pi$ if $P$ has the same value with the real transition kernel on the state-action pairs of the real trajectory from $s_e$ (or $(s_e, a_e)$ respectively) induced by $\pi$. This can always be achieved if $P$ is behaviour-accurate w.r.t. $\pi$.

We can take the above process as a mapping from transition kernels and policies to a function from $\mathcal{S}$ (or $\mathcal{S} \times \mathcal{A}$) to trajectories, and denote it as $O$. We can construct a set as $\{O(\widetilde{P}, \pi) \mid \pi \in \Pi\}$.

For any $M \in \mathcal{M}$, any $\xi(M)$-related function can be derived from one element from this class with behaviour-accuracy. $\qquad\square$

## E  THE DETAILED PROOF OF THEOREM 2

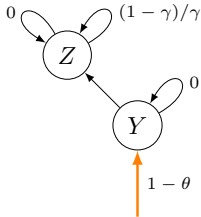

Figure 3: The changes in state $Y$. By selecting action 1, agents would transfer to state $Z$. Selecting actions in state $Z$ will only cause a self-loop, through which we can receive a reward $(1-\gamma)/\gamma$ with action 1 (resp. 2) for MDP in $\mathcal{M}_1$ or $\mathcal{M}_2$ (resp. $\mathcal{M}_3$ or $\mathcal{M}_4$). The dataset does not cover state $Z$.

Compared to the counterexamples in Theorem 1, we introduce the following changes:

1. we modify state $Y$ to introduce the insufficient data-coverage for selecting action 1 in state $Y$ (see Figure 3 for more details),

2. we introduce state $Z$ with variable rewards between MDPs,

3. the cardinalities of $I$ and $\bar{I}$ increase to $7|\mathcal{I}|/8$,

4. there are two transition kernels such that

$$
P_1(\cdot \mid s, a) = \begin{cases}
\text{Unif}(\mathcal{I}) & \text{if } s = \mathfrak{s} \text{ and } a = 1 \\
\text{Unif}(\bar{\mathcal{I}} \setminus \bar{I}) & \text{if } s = \mathfrak{s} \text{ and } a = 2 \\
7\text{Unif}(\{X\})/8 + \text{Unif}(\{Y\})/8 & \text{if } s \in \mathcal{I} \\
\text{Unif}(\{\bar{X}\}) & \text{if } s \in \bar{I} \\
\text{Unif}(\{Y\}) & \text{if } s \in \bar{\mathcal{I}} \setminus \bar{I} \\
\text{Unif}(\{Z\}) & \text{if } s = Y \text{ and } a = 1 \\
\text{Unif}(\{s\}) & \text{otherwise,}
\end{cases}
$$

$$
P_2(\cdot \mid s, a) = \begin{cases}
\text{Unif}(\mathcal{I} \setminus I) & \text{if } s = \mathfrak{s} \text{ and } a = 1 \\
\text{Unif}(\bar{\mathcal{I}}) & \text{if } s = \mathfrak{s} \text{ and } a = 2 \\
7\text{Unif}(\{X\})/8 + \text{Unif}(\{Y\})/8 & \text{if } s \in \bar{\mathcal{I}} \\
\text{Unif}(\{X\}) & \text{if } s \in I \\
\text{Unif}(\{Y\}) & \text{if } s \in \mathcal{I} \setminus I \\
\text{Unif}(\{Z\}) & \text{if } s = Y \text{ and } a = 1 \\
\text{Unif}(\{s\}) & \text{otherwise.}
\end{cases}
$$

5. there are two reward functions such that

$$
R_1(s, a) = \begin{cases}
1 - \gamma & \text{if } s \in \{X, \bar{X}\} \text{ and } a = 2 \\
(1-\gamma)/\gamma & \text{if } s = Z \text{ and } a = 1 \\
0 & \text{otherwise,}
\end{cases}
\qquad
R_2(s, a) = \begin{cases}
1 - \gamma & \text{if } s \in \{X, \bar{X}\} \text{ and } a = 2 \\
(1-\gamma)/\gamma & \text{if } s = Z \text{ and } a = 2 \\
0 & \text{otherwise.}
\end{cases}
$$

There are four MDP classes denoted as $\overline{\mathcal{M}} \coloneqq \{\mathcal{M}_1, \mathcal{M}_2, \mathcal{M}_3, \mathcal{M}_4\}$ and $\mathcal{M} = \mathcal{M}_1 \cup \mathcal{M}_2 \cup \mathcal{M}_3 \cup \mathcal{M}_4$. They share the same state space and action space, but differ in the following perspectives:

- the transition kernels for MDPs from $\mathcal{M}_1$ and $\mathcal{M}_2$ follow $P_1$, while those for MDPs from $\mathcal{M}_3$ and $\mathcal{M}_4$ follow $P_2$,

- the reward functions for MDPs from $\mathcal{M}_1$ and $\mathcal{M}_3$ follow $R_1$, while those for MDPs from $\mathcal{M}_2$ and $\mathcal{M}_4$ follow $R_2$.

This structure allows the optimal value function to be the same in $\mathcal{I}$ or $\overline{\mathcal{I}}$, breaking the barrier of $Q^\star$-realizability. However, as we can receive 1 return by selecting action 1 in state $Y$ with some policies, the dataset only covers suboptimal policies.

### E.1 FUNCITON CLASSES AND DATA DISTRIBUTION

**Policy function class** The policy function class $\Pi$ contains two policies $\pi_1$ and $\pi_2$, which are given by

$$\pi_1(s) = \begin{cases} 2 & \text{if } s = X \text{ or } s = Y \\ 1 & \text{otherwise,} \end{cases} \qquad \pi_2(s) = \begin{cases} 1 & \text{if } s = X \text{ or } s = Z \\ 2 & \text{otherwise.} \end{cases}$$

$\xi$ maps MDPs from $\mathcal{M}_1 \cup \mathcal{M}_2$ to $\pi_1$, and maps MDPs from $\mathcal{M}_3 \cup \mathcal{M}_4$ to $\pi_2$.

**Value function class** $\mathcal{Q}$ contains functions as summaried in Table 2.

Table 2: Functions in $\mathcal{Q}$.

| | $Q_{\pi_1}$ for $\mathcal{M}_1/\mathcal{M}_2$ | $Q_{\pi_2}$ for $\mathcal{M}_3/\mathcal{M}_4$ | $Q^\star$ for $\mathcal{M}_1/\mathcal{M}_2$ | $Q^\star$ for $\mathcal{M}_3/\mathcal{M}_4$ |
|---|---|---|---|---|
| $s = \mathfrak{s}$ & $a = 1$ | $7\gamma^2/8$ | $0$ | $\gamma^2$ | $\gamma^2$ |
| $s = \mathfrak{s}$ & $a = 2$ | $0$ | $7\gamma^2/8$ | $\gamma^2$ | $\gamma^2$ |
| $s \in \mathcal{I}$ | $7\gamma/8$ | $0$ | $\gamma$ | $\gamma$ |
| $s \in \overline{\mathcal{I}}$ | $0$ | $7\gamma/8$ | $\gamma$ | $\gamma$ |
| $s = X$ & $a = 1$ | $\gamma$ | $0$ | $\gamma$ | $\gamma$ |
| $s = X$ & $a = 2$ | $1$ | $1 - \gamma$ | $1$ | $1$ |
| $s = \overline{X}$ & $a = 1$ | $0$ | $\gamma$ | $\gamma$ | $\gamma$ |
| $s = \overline{X}$ & $a = 2$ | $1 - \gamma$ | $1$ | $1$ | $1$ |
| $s = Y$ & $a = 1$ | $0$ | $0$ | $1$ | $1$ |
| $s = Y$ & $a = 2$ | $0$ | $0$ | $\gamma$ | $\gamma$ |
| $s = Z$ & $a = 1$ | $1/\gamma - 1$ or $0$ | $1/\gamma - 1$ or $0$ | $1/\gamma$ or $1$ | $1$ or $\gamma/1$ |
| $s = Z$ & $a = 2$ | $0$ or $1/\gamma - 1$ | $0$ or $1/\gamma - 1$ | $1$ or $1/\gamma$ | $1/\gamma$ or $1$ |

**Partially accurate transition kernel** We set $\widetilde{P}$ as

$$\widetilde{P}(\cdot \mid s, a) = \begin{cases} \text{Unif}(\mathcal{I}) & \text{if } s = \mathfrak{s} \text{ and } a = 1 \\ \text{Unif}(\overline{\mathcal{I}}) & \text{if } s = \mathfrak{s} \text{ and } a = 2 \\ 7\text{Unif}(\{X\})/8 + \text{Unif}(\{Y\})/8 & \text{if } s \in \mathcal{I} \\ 7\text{Unif}(\{\overline{X}\})/8 + \text{Unif}(\{Y\})/8 & \text{if } s \in \overline{\mathcal{I}} \\ \text{Unif}(\{Z\}) & \text{if } s = Y \text{ and } a = 1 \\ \text{Unif}(\{s\}) & \text{otherwise.} \end{cases}$$

**State transition kernels with exploratory-accuracy** The set $\mathcal{P}_\mathcal{S}$ contains two elements described below:

$$P_{\mathcal{S},1}(\cdot \mid s) = \begin{cases} \text{Unif}(\mathcal{I}) & \text{if } s = \mathfrak{s} \\ 7\text{Unif}(\{X\})/8 + \text{Unif}(\{Y\})/8 & \text{if } s \in \mathcal{I} \\ \text{Unif}(\{Y\}) & \text{if } s \in \overline{\mathcal{I}} \\ \text{Unif}(\{s\}) & \text{otherwise,} \end{cases}$$

$$P_{\mathcal{S},2}(\cdot \mid s) = \begin{cases} \text{Unif}(\overline{\mathcal{I}}) & \text{if } s = \mathfrak{s} \\ 7\text{Unif}(\{\overline{X}\})/8 + \text{Unif}(\{Y\})/8 & \text{if } s \in \overline{\mathcal{I}} \\ \text{Unif}(\{Y\}) & \text{if } s \in \mathcal{I} \\ \text{Unif}(\{s\}) & \text{otherwise.} \end{cases}$$

**Reward function class**    We set $\mathcal{R}$ as $\{R_1, R_2\}$.

**Data distribution**    $d^{\mathcal{D}}$ follows that

$$d^{\mathcal{D}} = \frac{1}{2}\text{Unif}\Big(\{X, \overline{X}, Y, \mathfrak{s}\} \times \mathcal{A}\Big) + \frac{1}{2}\text{Unif}\Big((\mathcal{I} \cup \overline{\mathcal{I}}) \times \mathcal{A}\Big).$$

### E.2    PROOF SKETCH OF THEOREM 2

This subsection proves Theorem 2 through some high-level lemmas. The proof of these lemmas are deferred to the remainder of this section.

We denote the event of selecting action 1 in state $\mathfrak{s}$ as $E_1$, and the event of selecting action 1 in state $Z$ as $E_2$. For an algorithm $\mathfrak{A}$, policies behave suboptimally under MDP classes from $\overline{\mathcal{M}}$ if events from Table 3 happen (which we denote as $E_M^{\mathfrak{A}}$ for $M \in \mathcal{M}$) correspondingly. The exact value of this suboptimality follows Lemma 5.

**Lemma 5.** *Algorithm $\mathfrak{A}$ that outputs $\widehat{\pi}$ behaves suboptimally compared with $\xi(M)$ if $E_M^{\mathfrak{A}}$ happens, i.e., for all $M \in \mathcal{M}$,*

$$J_M(\xi(M)) - \mathbb{E}\big[J_M(\widehat{\pi})\big] \geq \gamma^2 \mathbb{P}(E_M^{\mathfrak{A}}) - \frac{\gamma^2}{8}. \tag{15}$$

Table 3: When policies behave suboptimally for MDPs in each MDP class. ✓: the event happens. ✗: the event does not happen.

|       | $\mathcal{M}_1$ | $\mathcal{M}_2$ | $\mathcal{M}_3$ | $\mathcal{M}_4$ |
|-------|-----------------|-----------------|-----------------|-----------------|
| $E_1$ | ✗ | ✗ | ✓ | ✓ |
| $E_2$ | ✓ | ✗ | ✗ | ✓ |

Constructing a distribution over MDP classes $\mathcal{M}$ as

$$d_{\mathcal{M}} := \text{Unif}(\mathcal{M}),$$

we can transfer the minimax lower bound into a Bayesian bound,

$$\inf_{\mathfrak{A}} \sup_{M} \Big[ J_M(\xi(M)) - \mathbb{E}\big[J_M(\widehat{\pi})\big] \Big] \geq \inf_{\mathfrak{A}} \mathbb{E}_{M \sim d_{\mathcal{M}}} \Big[ J_M(\xi(M)) - \mathbb{E}\big[J_M(\widehat{\pi})\big] \Big] \tag{16}$$

$$\geq \gamma^2 \inf_{\mathfrak{A}} \mathbb{E}_{M \sim d_{\mathcal{M}}} \Big[ \mathbb{P}(E_M^{\mathfrak{A}}) \Big] - \gamma^2/8. \tag{17}$$

We define $\mathbb{P}(E \mid \mathcal{M})$ as the averaged probability under $\mathcal{M}$ with sample size $N$, but omit $N$ for simplicity,

$$\mathbb{P}(E \mid \mathcal{M}) := \mathbb{E}_{M \sim \text{Unif}(\mathcal{M})}[\mathbb{P}_M^N(E)].$$

Eqn. (17) could be bounded with the total variation distance between some MDP families from $\overline{\mathcal{M}}$.

**Lemma 6.** *The fail probability is bounded by the total variation distances between certain MDP families from $\overline{\mathcal{M}}$.*

$$\inf_{\mathfrak{A}} \mathbb{E}_{M \sim d_{\mathcal{M}}} \Big[ \mathbb{P}(E_M^{\mathfrak{A}}) \Big] \geq \frac{1}{4} - \frac{1}{4} \max \Big\{ \text{TV}\Big(\mathbb{P}(\cdot|\mathcal{M}_1), \mathbb{P}(\cdot|\mathcal{M}_3)\Big), \text{TV}\Big(\mathbb{P}(\cdot|\mathcal{M}_1), \mathbb{P}(\cdot|\mathcal{M}_4)\Big),$$

$$\text{TV}\Big(\mathbb{P}(\cdot|\mathcal{M}_2), \mathbb{P}(\cdot|\mathcal{M}_3)\Big), \text{TV}\Big(\mathbb{P}(\cdot|\mathcal{M}_2), \mathbb{P}(\cdot|\mathcal{M}_4)\Big) \Big\}$$

$$- \frac{1}{2} \max \Big\{ \text{TV}\Big(\mathbb{P}(\cdot|\mathcal{M}_1), \mathbb{P}(\cdot|\mathcal{M}_2)\Big), \text{TV}\Big(\mathbb{P}(\cdot|\mathcal{M}_3), \mathbb{P}(\cdot|\mathcal{M}_4)\Big) \Big\}.$$

For notational convience, we abbreviate the total variation distance as

$$\mathrm{TV}(\mathcal{M}', \mathcal{M}'') := \mathrm{TV}\Big(\mathbb{P}(\cdot|\mathcal{M}'), \mathbb{P}(\cdot|\mathcal{M}'')\Big).$$

Intuitively, the difference in the reward function in state $Z$ does not matter since it is not covered by the dataset, and the difference in the transition kernel follows the same argument of Theorem 1. Thus, the total variation distance is small enough.

**Lemma 7.** *For any sample size $N$, we can choose the cardinality of $\mathcal{I}$ to ensure that the total variation distance is bounded between MDP classes from $\{\mathcal{M}_1, \mathcal{M}_2\}$ and MDP classes from $\{\mathcal{M}_3, \mathcal{M}_4\}$,*

$$\max_{\mathcal{M}' \in \{\mathcal{M}_1, \mathcal{M}_2\}, \mathcal{M}'' \in \{\mathcal{M}_3, \mathcal{M}_4\}} \mathrm{TV}(\mathcal{M}', \mathcal{M}'') \le 1/4.$$

**Lemma 8.** *The total variation distance between $\mathcal{M}_1$ and $\mathcal{M}_2$, as well as the one between $\mathcal{M}_3$ and $\mathcal{M}_4$, satisfies,*

$$\mathrm{TV}(\mathcal{M}_1, \mathcal{M}_2) = \mathrm{TV}(\mathcal{M}_3, \mathcal{M}_4) = 0.$$

Combining the above arguments yields that

$$\sup_M \inf_{\mathfrak{A}} \Big[ J_M(\xi(M)) - \mathbb{E}\big[J_M(\widehat{\pi})\big] \Big] \ge \gamma^2/16.$$

This completes the proof.

### E.3 PROOF OF LEMMA 5

**Lemma** (Restatement of Lemma 5). *Algorithm $\mathfrak{A}$ that outputs $\widehat{\pi}$ behaves suboptimally compared with $\xi(M)$ if $E_M^{\mathfrak{A}}$ happens, i.e., for all $M \in \mathcal{M}$,*

$$J_M(\xi(M)) - \mathbb{E}\big[J_M(\widehat{\pi})\big] \ge \gamma^2 \mathbb{P}(E_M^{\mathfrak{A}}) - \frac{\gamma^2}{8}. \tag{18}$$

*Proof.* If $E_M^{\mathfrak{A}}$ happens, $\widehat{\pi}$ can have 0 return in the best case. Since the optimal return from $\mathfrak{s}$ is $\gamma^2$, we have

$$\mathbb{E}_{E_M^{\mathfrak{A}}}\big[J_M(\widehat{\pi})\big] \le \gamma^2 \mathbb{P}(\overline{E}_M^{\mathfrak{A}}) \le \gamma^2 - \gamma^2 \mathbb{P}(E_M^{\mathfrak{A}}).$$

Thus, we have

$$\begin{aligned} J_M(\xi(M)) - \mathbb{E}\big[J_M(\widehat{\pi})\big] \ge & 7\gamma^2/8 - \gamma^2 + \gamma^2 \mathbb{P}(E_M^{\mathfrak{A}}) \\ \ge & \gamma^2 \mathbb{P}(E_M^{\mathfrak{A}}) - \frac{\gamma^2}{8}. \end{aligned}$$

This completes the proof. $\qquad\square$

### E.4 PROOF OF LEMMA 6

**Lemma** (Restatement of Lemma 6). *The fail probability is bounded by the total variation distances between certain MDP families from $\overline{\mathcal{M}}$.*

$$\begin{aligned} \inf_{\mathfrak{A}} \mathbb{E}_{M \sim d_{\mathcal{M}}}\big[\mathbb{P}(E_M^{\mathfrak{A}})\big] \ge & \frac{1}{4} - \frac{1}{4} \max\Big\{ \mathrm{TV}\Big(\mathbb{P}(\cdot|\mathcal{M}_1), \mathbb{P}(\cdot|\mathcal{M}_3)\Big), \mathrm{TV}\Big(\mathbb{P}(\cdot|\mathcal{M}_1), \mathbb{P}(\cdot|\mathcal{M}_4)\Big), \\ & \qquad\qquad \mathrm{TV}\Big(\mathbb{P}(\cdot|\mathcal{M}_2), \mathbb{P}(\cdot|\mathcal{M}_3)\Big), \mathrm{TV}\Big(\mathbb{P}(\cdot|\mathcal{M}_2), \mathbb{P}(\cdot|\mathcal{M}_4)\Big) \Big\} \\ & - \frac{1}{2} \max\Big\{ \mathrm{TV}\Big(\mathbb{P}(\cdot|\mathcal{M}_1), \mathbb{P}(\cdot|\mathcal{M}_2)\Big), \mathrm{TV}\Big(\mathbb{P}(\cdot|\mathcal{M}_3), \mathbb{P}(\cdot|\mathcal{M}_4)\Big) \Big\}. \end{aligned}$$

*Proof.* Due to the definition of $d_{\mathcal{M}}$,

$$
\begin{aligned}
&\inf_{\mathfrak{A}} \mathbb{E}_{M \sim d_{\mathcal{M}}} \left[ \mathbb{P}(E_M^{\mathfrak{A}}) \right] \\
=&\frac{1}{4} \inf_{\mathfrak{A}} \left[ \mathbb{P}(\overline{E}_1^{\mathfrak{A}} \text{ and } E_2^{\mathfrak{A}} | \mathcal{M}_1) + \mathbb{P}(\overline{E}_1^{\mathfrak{A}} \text{ and } \overline{E}_2^{\mathfrak{A}} | \mathcal{M}_2) + \mathbb{P}(E_1^{\mathfrak{A}} \text{ and } \overline{E}_2^{\mathfrak{A}} | \mathcal{M}_3) + \mathbb{P}(E_1^{\mathfrak{A}} \text{ and } E_2^{\mathfrak{A}} | \mathcal{M}_4) \right] \\
\geq&\frac{1}{4} \inf_{\mathfrak{A}} \Bigg[ \left[ \mathbb{P}(E_2^{\mathfrak{A}} | \overline{E}_1^{\mathfrak{A}}, \mathcal{M}_1) + \mathbb{P}(\overline{E}_2^{\mathfrak{A}} | \overline{E}_1^{\mathfrak{A}}, \mathcal{M}_2) \right] \cdot \min \left\{ \mathbb{P}(\overline{E}_1^{\mathfrak{A}} | \mathcal{M}_1), \mathbb{P}(\overline{E}_1^{\mathfrak{A}} | \mathcal{M}_2) \right\} \\
&+ \left[ \mathbb{P}(\overline{E}_2^{\mathfrak{A}} | \overline{E}_1^{\mathfrak{A}}, \mathcal{M}_3) + \mathbb{P}(E_2^{\mathfrak{A}} | \overline{E}_1^{\mathfrak{A}}, \mathcal{M}_4) \right] \cdot \min \left\{ \mathbb{P}(E_1^{\mathfrak{A}} | \mathcal{M}_3), \mathbb{P}(E_1^{\mathfrak{A}} | \mathcal{M}_4) \right\} \Bigg] \\
\geq&\frac{1}{4} \inf_{\mathfrak{A}} \Bigg[ \left[ 1 - \mathbf{TV}\Big( \mathbb{P}(\cdot | \overline{E}_1^{\mathfrak{A}}, \mathcal{M}_1), \mathbb{P}(\cdot | \overline{E}_1^{\mathfrak{A}}, \mathcal{M}_2) \Big) \right] \cdot \min \left\{ \mathbb{P}(\overline{E}_1^{\mathfrak{A}} | \mathcal{M}_1), \mathbb{P}(\overline{E}_1^{\mathfrak{A}} | \mathcal{M}_2) \right\} \\
&+ \left[ 1 - \mathbf{TV}\Big( \mathbb{P}(\cdot | \overline{E}_1^{\mathfrak{A}}, \mathcal{M}_3), \mathbb{P}(\cdot | \overline{E}_1^{\mathfrak{A}}, \mathcal{M}_4) \Big) \right] \cdot \min \left\{ \mathbb{P}(E_1^{\mathfrak{A}} | \mathcal{M}_3), \mathbb{P}(E_1^{\mathfrak{A}} | \mathcal{M}_4) \right\} \Bigg].
\end{aligned}
$$

We can use Lemma 2 to simplify the total variation distance of conditional probabilities,

$$
\begin{aligned}
&\inf_{\mathfrak{A}} \mathbb{E}_{M \sim d_{\mathcal{M}}} \left[ \mathbb{P}(E_M^{\mathfrak{A}}) \right] \\
\geq&\frac{1}{4} \inf_{\mathfrak{A}} \Bigg[ \min \left\{ \mathbb{P}(\overline{E}_1^{\mathfrak{A}} | \mathcal{M}_1), \mathbb{P}(\overline{E}_1^{\mathfrak{A}} | \mathcal{M}_2) \right\} - \mathbf{TV}\Big( \mathbb{P}(\cdot | \mathcal{M}_1), \mathbb{P}(\cdot | \mathcal{M}_2) \Big) \\
&+ \min \left\{ \mathbb{P}(E_1^{\mathfrak{A}} | \mathcal{M}_3), \mathbb{P}(E_1^{\mathfrak{A}} | \mathcal{M}_4) \right\} - \mathbf{TV}\Big( \mathbb{P}(\cdot | \mathcal{M}_3), \mathbb{P}(\cdot | \mathcal{M}_4) \Big) \Bigg] \\
\geq&\frac{1}{4} \inf_{\mathfrak{A}} \Bigg[ \min \Big\{ \mathbb{P}(\overline{E}_1^{\mathfrak{A}} | \mathcal{M}_1) + \mathbb{P}(E_1^{\mathfrak{A}} | \mathcal{M}_3), \mathbb{P}(\overline{E}_1^{\mathfrak{A}} | \mathcal{M}_1) + \mathbb{P}(E_1^{\mathfrak{A}} | \mathcal{M}_4), \\
&\qquad\qquad\qquad \mathbb{P}(\overline{E}_1^{\mathfrak{A}} | \mathcal{M}_2) + \mathbb{P}(E_1^{\mathfrak{A}} | \mathcal{M}_3), \mathbb{P}(\overline{E}_1^{\mathfrak{A}} | \mathcal{M}_2) + \mathbb{P}(E_1^{\mathfrak{A}} | \mathcal{M}_4) \Big\} \Bigg] \\
&- \frac{1}{2} \max \left\{ \mathbf{TV}\Big( \mathbb{P}(\cdot | \mathcal{M}_1), \mathbb{P}(\cdot | \mathcal{M}_2) \Big), \mathbf{TV}\Big( \mathbb{P}(\cdot | \mathcal{M}_3), \mathbb{P}(\cdot | \mathcal{M}_4) \Big) \right\}.
\end{aligned}
$$

The first term again can be transferred into the total variation distance,

$$
\begin{aligned}
&\inf_{\mathfrak{A}} \mathbb{E}_{M \sim d_{\mathcal{M}}} \left[ \mathbb{P}(E_M^{\mathfrak{A}}) \right] \\
\geq&\frac{1}{4} \inf_{\mathfrak{A}} \Bigg[ \min \Big\{ 1 - \mathbf{TV}\Big( \mathbb{P}(\cdot | \mathcal{M}_1), \mathbb{P}(\cdot | \mathcal{M}_3) \Big), 1 - \mathbf{TV}\Big( \mathbb{P}(\cdot | \mathcal{M}_1), \mathbb{P}(\cdot | \mathcal{M}_4) \Big), \\
&\qquad\qquad\qquad 1 - \mathbf{TV}\Big( \mathbb{P}(\cdot | \mathcal{M}_2), \mathbb{P}(\cdot | \mathcal{M}_3) \Big), 1 - \mathbf{TV}\Big( \mathbb{P}(\cdot | \mathcal{M}_2), \mathbb{P}(\cdot | \mathcal{M}_4) \Big) \Big\} \Bigg] \\
&- \frac{1}{2} \max \left\{ \mathbf{TV}\Big( \mathbb{P}(\cdot | \mathcal{M}_1), \mathbb{P}(\cdot | \mathcal{M}_2) \Big), \mathbf{TV}\Big( \mathbb{P}(\cdot | \mathcal{M}_3), \mathbb{P}(\cdot | \mathcal{M}_4) \Big) \right\}.
\end{aligned}
$$

Rearranging terms yields that

$$
\begin{aligned}
&\inf_{\mathfrak{A}} \mathbb{E}_{M \sim d_{\mathcal{M}}}\Big[\mathbb{P}(E_M^{\mathfrak{A}})\Big] \\
&\geq \frac{1}{4}\inf_{\mathfrak{A}}\bigg[1 - \max\Big\{\mathbf{TV}\Big(\mathbb{P}(\cdot|\mathcal{M}_1), \mathbb{P}(\cdot|\mathcal{M}_3)\Big), \mathbf{TV}\Big(\mathbb{P}(\cdot|\mathcal{M}_1), \mathbb{P}(\cdot|\mathcal{M}_4)\Big), \\
&\qquad\qquad\qquad\qquad \mathbf{TV}\Big(\mathbb{P}(\cdot|\mathcal{M}_2), \mathbb{P}(\cdot|\mathcal{M}_3)\Big), \mathbf{TV}\Big(\mathbb{P}(\cdot|\mathcal{M}_2), \mathbb{P}(\cdot|\mathcal{M}_4)\Big)\Big\}\bigg] \\
&\quad - \frac{1}{2}\max\Big\{\mathbf{TV}\Big(\mathbb{P}(\cdot|\mathcal{M}_1), \mathbb{P}(\cdot|\mathcal{M}_2)\Big), \mathbf{TV}\Big(\mathbb{P}(\cdot|\mathcal{M}_3), \mathbb{P}(\cdot|\mathcal{M}_4)\Big)\Big\} \\
&\geq \frac{1}{4} - \frac{1}{4}\max\Big\{\mathbf{TV}\Big(\mathbb{P}(\cdot|\mathcal{M}_1), \mathbb{P}(\cdot|\mathcal{M}_3)\Big), \mathbf{TV}\Big(\mathbb{P}(\cdot|\mathcal{M}_1), \mathbb{P}(\cdot|\mathcal{M}_4)\Big), \\
&\qquad\qquad\qquad\quad \mathbf{TV}\Big(\mathbb{P}(\cdot|\mathcal{M}_2), \mathbb{P}(\cdot|\mathcal{M}_3)\Big), \mathbf{TV}\Big(\mathbb{P}(\cdot|\mathcal{M}_2), \mathbb{P}(\cdot|\mathcal{M}_4)\Big)\Big\} \\
&\quad - \frac{1}{2}\max\Big\{\mathbf{TV}\Big(\mathbb{P}(\cdot|\mathcal{M}_1), \mathbb{P}(\cdot|\mathcal{M}_2)\Big), \mathbf{TV}\Big(\mathbb{P}(\cdot|\mathcal{M}_3), \mathbb{P}(\cdot|\mathcal{M}_4)\Big)\Big\}.
\end{aligned}
$$

This completes the proof. $\qquad\qquad\qquad\qquad\qquad\qquad\qquad\qquad\qquad\qquad\qquad\qquad$ $\square$

### E.5    PROOF OF LEMMA 7

**Lemma** (Restatement of Lemma 7). *For any sample size $N$, we can choose the cardinality of $\mathcal{I}$ to ensure that the total variation distance is bounded between MDP classes from $\{\mathcal{M}_1, \mathcal{M}_2\}$ and MDP classes from $\{\mathcal{M}_3, \mathcal{M}_4\}$,*

$$
\max_{\mathcal{M}' \in \{\mathcal{M}_1, \mathcal{M}_2\}, \mathcal{M}'' \in \{\mathcal{M}_3, \mathcal{M}_4\}} \mathbf{TV}(\mathcal{M}', \mathcal{M}'') \leq 1/4.
$$

*Proof.* Denote $\mathcal{M}' \in \{\mathcal{M}_1, \mathcal{M}_2\}$, and $\mathcal{M}'' \in \{\mathcal{M}_3, \mathcal{M}_4\}$, we first transfer the algorithm-dependent total variation distance into the dataset version through the data processing inequality,

$$
\begin{aligned}
\mathbf{TV}(\mathcal{M}', \mathcal{M}'') &= \mathbf{TV}\big(\mathbb{P}(\cdot|\mathcal{M}'), \mathbb{P}(\cdot|\mathcal{M}'')\big) \\
&\leq \mathbf{TV}_{\mathcal{D}}\Big(\mathbb{P}(\cdot \mid \mathcal{M}'), \mathbb{P}(\cdot \mid \mathcal{M}'')\Big).
\end{aligned}
$$

We take $\mathbb{P}(\cdot \mid \mathcal{M}')$ as $\mathbb{P}_1^N$ and take $P(\cdot \mid \mathcal{M}'')$ as $\mathbb{P}_2^N$. We then construct the dominating MDPs $M_0$ as done in Lemma 4, whose transition kernel follows that

$$
P_0(\cdot \mid s, a) := \begin{cases}
\mathrm{Unif}(\mathcal{I}) & \text{if } s = \mathfrak{s} \text{ and } a = 1 \\
\mathrm{Unif}(\overline{\mathcal{I}}) & \text{if } s = \mathfrak{s} \text{ and } a = 2 \\
7\mathrm{Unif}(\{X\})/8 + \mathrm{Unif}(\{Y\})/8 & \text{if } s \in \mathcal{I} \\
7\mathrm{Unif}(\{\overline{X}\})/8 + \mathrm{Unif}(\{Y\})/8 & \text{if } s \in \overline{\mathcal{I}} \\
\mathrm{Unif}(\{Z\}) & \text{if } s = Y \text{ and } a = 1 \\
\mathrm{Unif}(\{s\}) & \text{otherwise,}
\end{cases}
$$

and whose reward is $R_1$. We can follow exactly the same argument from Lemma 4 until Eqn. (13). By definition,

$$
\Phi_1(\theta|\overline{\mathcal{I}}|) = \frac{1/8 - (7/8 - \theta)}{(1/8)^2} = 64\theta - 48, \quad \Phi_2(\theta|\overline{\mathcal{I}}|) = \frac{7}{8} \cdot \frac{\theta}{(7/8)^2} + (1 - 7/8)\frac{1/8 - (7/8 - \theta)}{(1 - 7/8)^2} = \frac{64\theta}{7} - 6.
$$

Both of them are monotonic.

Since $T \sim \text{Hyper}(7|\bar{\mathcal{I}}|/8, |\bar{\mathcal{I}}|, 7|\bar{\mathcal{I}}|/8)$,

$$\mathbb{E}_{T \sim \text{Hyper}(7|\bar{\mathcal{I}}|/8, |\bar{\mathcal{I}}|, 7|\bar{\mathcal{I}}|/8)}\left[\left(\frac{11}{16} + \frac{1}{16}\Phi_1(T) + \frac{1}{4}\Phi_2(T)\right)^N\right]$$

$$= \sum_{t=0}^{7|\bar{\mathcal{I}}|/8} \frac{\binom{7|\bar{\mathcal{I}}|/8}{t}\binom{|\bar{\mathcal{I}}|/8}{7|\bar{\mathcal{I}}|/8-t}}{\binom{|\bar{\mathcal{I}}|}{7|\bar{\mathcal{I}}|/8}}\left(\frac{11}{16} + \frac{1}{16}\Phi_1(t) + \frac{1}{4}\Phi_2(t)\right)^N$$

$$= \sum_{t=0}^{7(7/8+\epsilon)|\bar{\mathcal{I}}|/8} \frac{\binom{7|\bar{\mathcal{I}}|/8}{t}\binom{|\bar{\mathcal{I}}|/8}{7|\bar{\mathcal{I}}|/8-t}}{\binom{|\bar{\mathcal{I}}|}{7|\bar{\mathcal{I}}|/8}}\left(\frac{11}{16} + \frac{1}{16}\Phi_1(t) + \frac{1}{4}\Phi_2(t)\right)^N$$

$$+ \sum_{t=(7/8+\epsilon)|\bar{\mathcal{I}}|/8}^{7|\bar{\mathcal{I}}|/8} \frac{\binom{7|\bar{\mathcal{I}}|/8}{t}\binom{|\bar{\mathcal{I}}|/8}{7|\bar{\mathcal{I}}|/8-t}}{\binom{|\bar{\mathcal{I}}|}{7|\bar{\mathcal{I}}|/8}}\left(\frac{11}{16} + \frac{1}{16}\Phi_1(t) + \frac{1}{4}\Phi_2(t)\right)^N \qquad (0 \le \epsilon \le \tfrac{49}{64}|\bar{\mathcal{I}}|)$$

$$\le \underbrace{\left(\frac{11}{16} + \frac{1}{16}\Phi_1(7(7/8+\epsilon)|\bar{\mathcal{I}}|/8) + \frac{1}{4}\Phi_2(7(7/8+\epsilon)|\bar{\mathcal{I}}|/8)\right)^N}_{(1)}$$

$$+ \underbrace{\mathbb{P}(T \ge 7(7/8+\epsilon)|\bar{\mathcal{I}}|/8)\left(\frac{11}{16} + \frac{1}{16}\Phi_1(7|\bar{\mathcal{I}}|/8) + \frac{1}{4}\Phi_2(7|\bar{\mathcal{I}}|/8)\right)^N}_{(2)}.$$

For term $(1)$, we can rewrite it as

$$\left(\frac{11}{16} + \frac{1}{16}\Phi_1(7(7/8+\epsilon)|\bar{\mathcal{I}}|/8) + \frac{1}{4}\Phi_2(7(7/8+\epsilon)|\bar{\mathcal{I}}|/8)\right)^N$$

$$= \left(\frac{11}{16} + \frac{1}{16}\cdot 64 \cdot \left(\frac{49}{64} + \frac{7\epsilon}{8}\right) - 3 + \frac{1}{4}\cdot\left(\frac{64}{7}\cdot 7(7/8+\epsilon)/8 - 6\right)\right)^N$$

$$= \left(1 + \frac{7\epsilon}{2}\right)^N.$$

For term $(2)$, we rewrite its latter part as

$$\left(\frac{11}{16} + \frac{1}{16}\Phi_1(7|\bar{\mathcal{I}}|/8) + \frac{1}{4}\Phi_2(7|\bar{\mathcal{I}}|/8)\right)^N$$

$$= \left(\frac{11}{16} + \frac{1}{16}\cdot(56-48) + \frac{1}{4}\cdot\left(\frac{64}{7}\cdot\frac{7}{8} - 6\right)\right)^N$$

$$= \left(\frac{27}{16}\right)^N.$$

The probability part of term $(2)$ is exponentially small because of the hypergeometric tail bound (Lemma 1), which yields that

$$\mathbb{P}(T \ge 7(7/8+\epsilon)|\bar{\mathcal{I}}|/8)\left(\frac{27}{16}\right)^N$$

$$\le \exp(-2\epsilon^2 \cdot 7|\bar{\mathcal{I}}|/8)\left(\frac{27}{16}\right)^N$$

$$= \exp\left(-\frac{7}{4}\epsilon^2|\bar{\mathcal{I}}| + N\log\left(\frac{27}{16}\right)\right).$$

Thus, the total variation distance is bounded as

$$\mathrm{TV}_{\mathcal{D}}\Big(\mathbb{P}_1^N, \mathbb{P}_0^N\Big) \leq \frac{1}{2}\sqrt{\left(1+\frac{7\epsilon}{2}\right)^N + \exp\left(-\frac{7}{4}\epsilon^2|\mathcal{I}| + N\log\left(\frac{27}{16}\right)\right) - 1}$$

$$\leq \frac{1}{2}\sqrt{\exp\left(\frac{7\epsilon N}{2}\right) + \exp\left(-\frac{7}{4}\epsilon^2|\mathcal{I}| + N\log\left(\frac{27}{16}\right)\right) - 1}.$$

Taking $0 < \epsilon \leq \frac{2}{7N}\log(33/32)$ and $|\bar{\mathcal{I}}| = |\mathcal{I}| \geq \frac{4\left(N\log\left(\frac{27}{16}\right)+\log 32\right)}{7\epsilon^2}$ yields that

$$\mathrm{TV}_{\mathcal{D}}(\mathbb{P}(\cdot|\mathcal{M}'), \mathbb{P}_0^N) \leq 1/8.$$

Following the same argument, we also have

$$\mathrm{TV}_{\mathcal{D}}(\mathbb{P}(\cdot|\mathcal{M}''), \mathbb{P}_0^N) \leq 1/8.$$

Therefore,

$$\max_{\mathcal{M}'\in\{\mathcal{M}_1,\mathcal{M}_2\},\mathcal{M}''\in\{\mathcal{M}_3,\mathcal{M}_4\}} \mathrm{TV}(\mathcal{M}',\mathcal{M}'') \leq \mathrm{TV}_{\mathcal{D}}(\mathbb{P}(\cdot|\mathcal{M}'),\mathbb{P}_0^N) + \mathrm{TV}_{\mathcal{D}}(\mathbb{P}(\cdot|\mathcal{M}''),\mathbb{P}_0^N)$$

$$\leq 1/4.$$

This completes the proof. □

### E.6 PROOF OF LEMMA 8

**Lemma** (Restatement of Lemma 8). *The total variation distance between $\mathcal{M}_1$ and $\mathcal{M}_2$, as well as the one between $\mathcal{M}_3$ and $\mathcal{M}_4$, satisfies,*

$$\mathrm{TV}(\mathcal{M}_1,\mathcal{M}_2) = \mathrm{TV}(\mathcal{M}_3,\mathcal{M}_4) = 0.$$

*Proof.* We can first transfer the total variation distance between $\mathcal{M}_1$ and $\mathcal{M}_2$ into the total variation distance of the dataset via data processing inequality,

$$\mathrm{TV}(\mathcal{M}_1,\mathcal{M}_2) \leq \mathrm{TV}_{\mathcal{D}}\Big(\mathbb{P}(\cdot \mid \mathcal{M}_1), \mathbb{P}(\cdot \mid \mathcal{M}_2)\Big).$$

Under the data distribution $d^{\mathcal{D}}$, the dataset generating principle of $\mathcal{M}_1$ and $\mathcal{M}_2$ is exactly the same since their different part is not covered by $d^{\mathcal{D}}$. This means that

$$\mathrm{TV}_{\mathcal{D}}\Big(\mathbb{P}(\cdot \mid \mathcal{M}_1), \mathbb{P}(\cdot \mid \mathcal{M}_2)\Big) = 0.$$

One can also prove $\mathrm{TV}(\mathcal{M}_3,\mathcal{M}_4) = 0$ in a similar manner. □

## F COMPARING THE ASSUMPTIONS FROM THIS PAPER AND PRIOR WORKS

This section reviews assumptions in offline RL with general function approximation and presents a comparison of these assumptions across different studies (including ours). Note that our results are lower bounds so the strength of assumptions should be viewed inversely in some cases. We first elaborate data assumptions in Appendix F.1, and then function assumptions in Appendix F.2. A summary of their usage in the literature can be found in Table 4.

A worth-mentioning point is that model-based assumptions are aligned with the toy assumptions (such as tabular MDPs), since they assume that the possible underlying models are "small." Model realizability implies most function class assumptions. For the stringency of model realizability, we do not discuss the model-based assumptions that appears in other works (Uehara & Sun, 2021; Bhardwaj et al., 2023) in this section,

Table 4: Summary of major assumptions in Offline RL with general function approximation. Realizability-type means that the results in the papers only require realizability-type assumptions, and completeness-type means that the results in the papers make completeness-type assumptions. The rows summarize the data assumptions made in the papers. We mark papers with model-based assumptions with $M$ superscript.

| | Realizability-type | Completeness-type |
|---|---|---|
| Single Coverage | Theorem 2 (ours); Zhan et al. (2022); Uehara & Sun (2021)$^M$; Bhardwaj et al. (2023)$^M$ | Liu et al. (2020); Xie & Jiang (2020b); Xie et al. (2021a); Cheng et al. (2022); Rashidinejad et al. (2022); Zhu et al. (2023) |
| Exploratory Coverage | | Chen & Jiang (2019) |
| The data assumptions made by Xie & Jiang (2020a) | Theorem 1 (ours); Xie & Jiang (2020a) | |

## F.1 Data Assumptions

Data assumption are the cornerstones of offline RL—they essentially determine what we can access as empirical facts. When comparing to online RL, data assumptions are always considered as an alternative of the frameworks in general function approximation (Xie et al., 2022). As discussed in Chen & Jiang (2019), data assumptions can also be viewed as implicit model assumptions.

As briefly introduced in Section 2.2, bounds on the coverage coefficient are the main stream of data assumptions. An interesting alternative for the ratio of distributions is to involve loss functions, as done in works such as Xie et al. (2021a); Cheng et al. (2022); Bhardwaj et al. (2023). This approach takes the form of

$$C := \max_{f \in \mathcal{F}} \frac{\mathbb{E}_\nu\big[\mathcal{L}(f)\big]}{\mathbb{E}_{d^\mathcal{D}}\big[\mathcal{L}(f)\big]}, \tag{19}$$

where $\nu$ is the distribution we want to shift to (e.g., the induced distribution of the optimal policy), $\mathcal{F}$ is the function class, $\mathcal{L}$ is the loss function in the algorithm, and $C$ is the refined concentration coefficient. This refined notion of coverage is yet no-surprising since in most cases we only use the coverage assumptions for loss-shifting, or namely, distribution rescaling.

**Concentration w.r.t.** $P$ **and** $\mathcal{A}$ The coverage assumptions in Xie & Jiang (2020a) require that

- $\mu^\mathcal{D}$ (the state margin of $d^\mathcal{D}$) should scale at the same level as the transition kernel $P(\cdot|s, a)$ and the initial state distribution, namely, there exists a constant $C > 0$ such that for all $s, s' \in \mathcal{S}, a \in \mathcal{A}$,

$$P(s'|s, a)/\mu^\mathcal{D}(s') \leq C \quad \text{and} \quad \mu_0(s)/\mu^\mathcal{D}(s) \leq C.$$

- for every state $s \in \mathcal{S}$, the behavior policy $\pi_b(\cdot|s)$ should be lower-bounded for all actions $a \in \mathcal{A}$ (thus covers all possible actions).

These assumptions are remarkably strong, exploiting the access of the dataset and making strong restrictions on the possible MDPs. Conditions 2 and 3 in Theorem 1 are specifications of these assumptions, and the data assumptions in Theorem 1 are thus stronger than the ones in Xie & Jiang (2020a).

One can also verify that the data assumptions in Theorem 1 are stronger than almost all those made in offline RL literature (e.g., Chen & Jiang (2019); Xie & Jiang (2020b); Xie et al. (2021a); Cheng et al. (2022); Ozdaglar et al. (2022); Zhu et al. (2023); Uehara et al. (2023)).

## F.2 Function Assumptions

Function assumptions in offline RL can be approximately characterized in three aspects: the loss to be minimized, the number of loss presented in the assumption, and the complexity of the function class. The last aspect is used to analysis the generalization property of the function class, and its treatment in offline RL does not differ from its use in the one in empirical processes (Pollard, 1990; Vaart & Wellner, 1996). We therefore only elaborate the first two aspects in the reminder of this

section. A summary of function assumptions is presented in Table 5. It is worth mentioning that this treatment is primarily aimed at real-world applications. Much of the analysis in RL still focuses on simpler scenarios such as tabular MDPs and linear MDPs, in which cases most function assumptions are naturally met, since the university of all possible functions is simple enough.

### F.2.1 On the Loss to Be Minimized

Functions classes, or approximators, are designed to minimize losses, and the quality of a function class can be determined by the scale of the minimum loss of the function class. We would like to mention that, the assumptions described below can often be converted to weaker ones by modifying the proofs without loss of generality.

**Strictly Containing** One may assume that the function exactly contains certain desired elements, or can optimally minimize a loss function. For instance, we may assume a value function class containing the optimal value functions, or equally, have a function minimize the Bellman optimal equation. From the perspective of loss, we can interpret it as having a function class $\mathcal{F}$ satisfying

$$\min_{f \in \mathcal{F}} \underbrace{\|f - f^\star\|_\infty}_{\text{the loss}} = 0.$$

This approach has more stringent requirements but also offers greater flexibility in application, possibly easing the proof process. Taking the optimal value function as an example, we can use the function class to minimize the optimal Bellman error, as well as to serve as a discriminator for the density ratio in MIS.

**Approximately Containing** A refined approach is to measure the violation of an assumption by an constant and then state the final result as a function of this constant. For instance, we may assume having a function class $\mathcal{F}$ satisfying

$$\min_{f \in \mathcal{F}} \underbrace{\|f - f^\star\|_{\infty, d^{\mathcal{D}}}}_{\text{the loss}} \le \epsilon,$$

by including a constant $\epsilon > 0$ to measure the sub-optimality. We can also replace the loss above with more refined ones such as the Bellman error (Zhu et al., 2023).

### F.2.2 On the Number of Losses

The number of losses required to be minimized measures another aspect of function assumptions. Having exponentially many losses to minimize versus having just a few are completely different, which forms the fundamental difference between realizability-type assumptions and completeness-type assumptions. In real-world applications, completeness-type assumptions may not always be met, leading to instability in the training process that could potentially destroy the final result.

**Realizability-type** Realizability-type assumptions are the assumptions that there are only a few losses to minimize. For example, we may assume that we have a function class $\mathcal{F}$ such that

$$\max\{\min_{f \in \mathcal{F}} \mathcal{L}_1(f), \min_{f \in \mathcal{F}} \mathcal{L}_2(f)\} = 0,$$

where $\mathcal{L}_1$ and $\mathcal{L}_2$ are two loss functions. These assumptions are natural extension of the ones in supervised learning, and are typically considered standard and relatively mild.

**Completeness-type** Completeness-type assumptions are the assumptions in which the number of objective losses are exponentially many, or are approximately the same as the cardinality of a function class. For example, we may assume that we have a function class $\mathcal{F}$ such that

$$\max_{f \in \mathcal{F}} \min_{f' \in \mathcal{F}} \mathcal{L}(f, f') = 0,$$

where $\mathcal{L} : \mathcal{F} \times \mathcal{F} \to \mathbb{R}$ is a certain loss function, e.g., the Bellman residual,

$$\mathcal{L}(f, f') \coloneqq \|f - \mathcal{T}f'\|_{2, d^{\mathcal{D}}}.$$

These assumptions arise from the requirements of iterative algorithms such as FQI. Concretely, taking FQI as an example, let $\mathcal{F} \subseteq \mathcal{S} \times \mathcal{A} \to \mathbb{R}$ be the value function class. In each step of the algorithm, denoted as $t$, we assign a new function $f_{t+1}$ based on the previous one $f_t$, as

$$f_{t+1} = \min_{f \in \mathcal{F}} \|f - \mathcal{T}f_t\|_{2, d^{\mathcal{D}}}. \tag{20}$$

FQI aims for the assignment to follow $f_{t+1} = \mathcal{T} f_t$, which corresponds to the Bellman update. The assumption above is famously addressed as the Bellman completeness.

Table 5: Summary of the Function Assumptions in the offline RL literature. Realizability-type means that the results in the papers only require realizability-type assumptions, and completeness-type means that the results in the papers make completeness-type assumptions. The rows correspond to two different types of losses used in the assumption. The superscript '+' means that there are some accompanying assumptions (e.g., the gap assumption (Chen & Jiang, 2022)) for the function classes in this paper. We mark papers with model-based assumptions with a superscript 'M'.

| | Realizability-type | Completeness-type |
|---|---|---|
| Strictly Containing | Uehara & Sun (2021)$^M$; Chen & Jiang (2022)$^+$;Uehara et al. (2023)$^+$; Bhardwaj et al. (2023)$^M$; Zhan et al. (2022); Theorem 1 (ours) | |
| Approximated Containing | Chen & Jiang (2019); Rashidinejad et al. (2022); Theorem 2 (ours) | Liu et al. (2020); Xie et al. (2021a); Cheng et al. (2022); Zhu et al. (2023); Xie & Jiang (2020b) |

### F.2.3 COMPARING THE FUNCTION ASSUMPTIONS WITH OURS

Our function class assumptions in Theorem 1 are weaker than those made in model-based analyses (Uehara & Sun, 2021). When specifying to concrete functions such as the value function and the density ratio, one can show that our function class assumptions are weaker than those with completeness-type assumptions (Chen & Jiang, 2019; Liu et al., 2020; Xie et al., 2021a; Rashidinejad et al., 2022), and are comparable (and sometimes the same) with the works that make only realizability-type assumptions (Xie & Jiang, 2020a; Zhan et al., 2022). We aim to demonstrate that completeness-type assumptions are particularly challenging to be mitigated.

