# OpenReview forum: "On the Role of General Function Approximation in Offline Reinforcement Learning"
_ICLR.cc/2024/Conference — ICLR 2024 spotlight_

### Official Review · Reviewer_MhTC · 2023-10-30

**Soundness:** 3 good
**Presentation:** 2 fair
**Contribution:** 3 good
**Rating:** 8
**Confidence:** 4

**Summary:**

This paper studies the offline RL with general function approximation.

The central problem of this topic is to identify the structural assumptions that empower sample-efficient learning. Many previous works impose assumptions on the data coverage and/or assumptions on the function class/MDP structures. Recently, foster et. al. established the fundamental limit when the completeness is missing but there are also works like [1] and [2], bypassing the lower bound in foster et. al. with different function approximation target.

This work aims  to present a rather unified framework to connect the function approximation to the realizability, which is interesting and very relevant to the community.


[1] Offline reinforcement learning with realizability and single-policy concentrability
[2] When is realizability sufficient for off-policy reinforcement learning?

**Strengths:**

The topic is interesting. I have been curious about what we can do without strong completeness-type assumption and super-strong data coverage conditions since [1]. While there are works bypassing the lower bounds in [1] with different type of realizability assumptions and refined characterization of completeness [2, 3], the relationship between various assumptions in the literature is still not very clear. So it is exciting to see that the authors are trying to fill the gap.

However, the current version is still not ready for publication due to the writing quality and since the relationships to previous works are not clear.

[1] Offline reinforcement learning: Fundamental barriers for value function approximation
[2] Offline reinforcement learning with realizability and single-policy concentrability
[3] When is realizability sufficient for off-policy reinforcement learning?

**Weaknesses:**

The main drawback of this paper is the writing. The authors state many things without concrete examples and arguments but only abstract summarizations (by using the word certain). For instance, the authors mention the previous works of ``Xie and Jiang''by their refined data assumptions, which assuming that the audience is familiar with these previous works. Meanwhile, many notations are used without a formal definitions although I can understand some of them as they have been used by the previous papers but this can still be a problem for general readers.

1 It would be great if the authors could instantiate the definitions present in this paper or discuss the relationships with the previous assumptions like the Bellman completeness in [1] and the realizability in [2]. For instance, in the example 2, it would be better to prove that the standard Q-pi-realizability is a special case of definition 2 by explicitly specifying the $\mathcal{F}$ and $\mathcal{G}$.

2 In example 3, may I take \phi as the mapping from a Q-function to its greedy policy? So completeness w.r.t. $\mathcal{T}^\pi$ for all the functions whose greedy policy is $\pi$ is indeed completeness w.r.t. $\mathcal{T}$ for all the functions in $\mathcal{F}$?

3 While proposition 1 is intuitive and reasonable, it seems that many previous works are indeed constructing the lower bounds through a worst-case consideration on the MDP by constructing a family of MDP classes satisfying the assumptions (e.g. [3]).

4 How did the conditions of theorem 1 compare to the assumptions in [1,2]? This should be highlighted so that the readers can evaluate the roles of the corollaries presented in section 5.1.

5 Most of the coverage assumptions are made with respect to the $\xi(M) \in \Pi$, which is more related to the coverage over $\Pi$, in contrast to the single-policy coverage assumption ($\pi^*$) commonly used in the previous works. I am trying to understand the case of $\pi^* \in \Pi$. But due to the presence of $\xi$, the obtained lower bound may not compete with $\pi^*$. Can you give an example of the unknown mapping $\xi(\cdot)$ so that we can understand the meaning of the target $J_M(\xi(M))-J_M(\hat{\pi})$?



[1] Bellman-consistent Pessimism for Offline Reinforcement Learning

[2] Offline reinforcement learning with realizability and single-policy concentrability

[3] is pessimism provably efficient for offline rl

**Questions:**

See weakness.

---

> ### Author Response · Authors · 2023-11-14
>
> We appreciate the reviewer for the valuable feedback, particularly the critical comments on writing have helped us improve the presentation of our paper. We have addressed the comments, and incorporated your suggestions in the revised paper. We hope the reviewer will be content with our response and revision. Additional comments are also welcome during the discussion phase.
>
> ## Response to the first paragraph in the weakness part:
>
> Your comments have been well-taken. We acknowledge that the paper and certain terminologies are somewhat technical. One reason for this perception is that we intended to convey as much information as possible to readers, which may result in a lack of clarity for general readers. In fact, to address this, we included a notation table in Appendix A to ensure the clarity of the notations.
>
> In response to your comment about the data assumption by Xie and Jiang, we have added a detailed description at its first occurrence in the revision. Moreover, we have also incorporated your other suggestions into the revision (please refer to our subsequent responses). We hope that the presentation of our revised paper meets your expectations.
>
> ## Response to Weakness 1
>
> Thanks for your suggestion. The definition of Bellman-completeness was provided in Example 3. In the revision, we have added its definition at its initial occurrence to ensure better clarity.
> For the comparison with assumptions in previous works, please see our response to weakness 4.
> Regarding the Q-pi-realizability in example 2, corresponding to Definition 2, we have $\mathcal{F}^\star=${$Q_{\pi}|\pi\in\Pi$}, $\mathcal{F}=\mathcal{Q}$ and $\mathcal{G}=\Pi$.  Explicit specifications in the examples have also been added in the revision.
>
> ## Response to Weakness 2
>
> Yes, you are correct. As stated in Example 3, the function $\phi$ is a mapping from $q$ to $\pi_q$, where $\pi_q$ is the optimal/greedy policy for $q$. This conversion is to emphasize the role of policy in RL analysis.
>
> ## Response to Weakness 3
>
> Most previous works construct MDP families satisfying assumptions of model-free functions (as mentioned in section 2.4.), whereas we propose building lower bounds for model-based functions.
> This type of construction enables us to extend the lower bound to a wide class of functions, making it more generic and versatile.
>
> ## Response to Weakness 4
>
> As stated in the paragraph before theorem 1, our data assumptions are stronger than the partial coverage (the assumption in works like [1,2,4,8]), exploratory coverage (the assumptions in works like [7]), and some refined data assumptions [5,6].
>
> Our function assumptions are designed to verify if learning with weak function assumptions is possible.
> To build some generic lower bounds, we focus on the model class in the theorem.
> Our function class assumptions are weaker than the ones made in model-based analysis ([8], as mentioned in remark 4).
> A comparison with model-free algorithms is hard to compose.
> Nevertheless, when specifying to concrete functions like the value function and the density ratio,
> one can show that our function class assumptions are weaker than the ones with completeness-type assumptions (like [1,4,7]),
> and are at the same level compared with literature making only realizability-type assumptions (like [2,5,6]).
> We aim to demonstrate that completeness-type assumptions are particularly challenging to be mitigated.
> We have added this part of the comparison to our paper (Appendix F, due to the page limitation). Thanks again for your advice.
>
> ## Response to Weakness 5
>
> The lower bound construction (as shown in section D.1) can actually ensure that we can have $J_M(\xi(M))=J_M(\pi^\star)$, i.e., $\xi(M)$ is optimal w.r.t. $M$ from the initial distribution.
> Therefore, we have $J_M(\xi(M))-J_M(\hat{\pi})=J_M(\pi^\star_M)-J_M(\hat{\pi})$.
> This makes the comparison interesting.
> Also, a relationship with the coverage assumptions in previous works is that: coverage assumptions w.r.t. $\xi(M)$ are the same as $\pi^\star$ (which is optimal from every state or state-action pair), since we only consider coverage assumptions from the initial distribution.
>
> Moreover, as the dataset is uncontrollable in most cases, assuming it covers the optimal policy is a bit strong.
> One may instead want to learn a policy that performs better than a specific one. This corresponds to the use of $\xi$.
>
> [1] Bellman-consistent Pessimism for Offline Reinforcement Learning
>
> [2] Offline reinforcement learning with realizability and single-policy concentrability
>
> [3] is pessimism provably efficient for offline rl
>
> [4] Provably good batch reinforcement learning without great exploration
>
> [5] Batch value-function approximation with only realizability
>
> [6] Refined value-based offline rl under realizability and partial coverage
>
> [7] information-theoretic considerations in batch reinforcement learning
>
> [8] Pessimistic Model-based Offline Reinforcement Learning under Partial Coverage

---

> > ### Comment · Reviewer_MhTC · 2023-11-18
> > **Thanks for the detailed responses**
> >
> > Thanks for the detailed responses.
> >
> > I believe that this is an interesting and solid works, and is also novel in providing new insights for the readers. The only concern from my side is the clarity and the way of conveying the ideas and information. The responses are clear and address most of my concerns so I raise my score to 6 to support the acceptance of this work. I still have some minor problems and suggestions.
> >
> > 1. It is a good idea to use one section (in the appendix) to throughly review the existing function approximation assumption and data coverage assumption, where you may clearly state their formal definition and the intuitions, followed by some comparisons to reveal their connections and differences. I believe that this can not only make the paper more friendly to the readers, but also promote the impacts of the work. You may achieve this goal by elaborating on the last section in the revised version.
> >
> > 2 In the last section, you state that the realizability-type assumption in theorem 2 is comparable with Zhan et al. Does this mean the two assumptions can imply each other? Or does it mean they are only similar in principle (realizability)? Could you position the result of Zhan at al. 2022 in the unified framework of this paper? In my mind, Zhan et al. 2022 considers a different function approximation method (compared to the value-based one in the Foster et al. 2021). Since we have a unified lower bound here, an interpretation of this less-explored function approximation method would be interesting and showcase the power of the presented framework.
> >
> > 3 I agree that in most of times, we may not expect that the dataset covers the pi^* well but only can expect to compete with the best policy covered by the offline dataset. I would like to thank for the classification on $J_M(\xi(M))=J_M(\pi^\star)$ and this makes a lot of sense. But I noticed that in [1] (and also some other works like [2] where the results can be easily extended to compete with the covered policies with minor modification), the results are indeed with respect to any policies (in a pre-determined policy class though). Therefore, it is still helpful to give a concrete example of \xi to help the readers to evaluate the theorems provided in this paper.
> >
> > Thanks again for your efforts in exploring this core problems in rl theory.
> >
> > [1] Bellman-consistent Pessimism for Offline Reinforcement Learning
> > [2] is pessimism provably efficient for offline rl

---

> > > ### Author Response · Authors · 2023-11-20
> > >
> > > We would like to thank the reviewer for the positive feedback. We also appreciate your new comments and suggestions regarding our work.
> > >
> > > ## Response to Point 1
> > >
> > > This suggestion is definitely meaningful. In the future final vision (should the paper be accepted), we will comprehensively enhance Appendix F by providing thorough review of the function and data assumptions made in prior works. We will formally introduce their definitions and intuitions, and compare the assumptions in different works in greater details. Additionally, we will expand Table 4 to offer a clearer comparison among different works. Thanks again for this helpful suggestion, which will undoubtedly contribute to enhancing the clarity of our paper.
> > >
> > > ## Response to Point 2
> > >
> > > To state first, as we study lower bounds and Zhan et al. 2022 studies upper bounds, it is inevitable that there are some gaps in our assumptions (note that we both consider the problem of whether learning a good policy is possible, rather than deriving a concrete sample complexity bound, etc.).
> > > Although our function assumptions follow the same principle, our results cannot imply each other.
> > >
> > > We prefer comparing Zhan et al. 2022 with Theorem 1 and Corollary 1 in our paper (as the function assumption in our Corollary 1 is closer to that in Zhan et al). We focus on Corollary 1 in their paper
> > > since it compares the learned policy with the optimal policy, while their Theorem 1 only compares the learned policy with the regularized policy.
> > >
> > > From the dataset perspective, Zhan et al. 2022 makes coverage assumptions w.r.t.
> > > a regularized optimal policy and the optimal policy in Corollary 1.
> > > Our data assumptions cover theirs.
> > >
> > > From the function perspective, Zhan et al. 2022 combines marginalized importance sampling (MIS) with linear programming (LP) methods in MDPs. The additional functions used are density ratios, for which they require the realizability of the density ratio of the (regularized) optimal policy. We discuss the density ratio in our Corollary 1 as well.
> > > Meanwhile, Zhan et al. 2022 requires the realizability of the (regularized) optimal value function, which is excluded in Theorem 1 as stated in section 5.2 (their regularization tends to vanish in corollary 1 in their paper).
> > > If we could take $\xi(M)$ as the optimal policy $\pi^\star$ (while it's not possible in reality), assumptions in Corollary 1 would be nearly the same as those in Zhan et al. 2022.
> > > Moreover, we argue that our results are interesting since the function and coverage assumptions made by Zhan et al. 2022 are overly specific for two policies---the optimal one and the regularized optimal one.
> > > These might make their results rarely applicable in practical scenarios.
> > >
> > > The reviewer mentioned that "Since we have a unified lower bound here, an interpretation of this less-explored function approximation method would be interesting and showcase the power of the presented framework."
> > > This is indeed a strength of our work. Based on the generic lower bound in Theorem 1, we are able to derive a more specific lower bound in Corollary 1 that incorporates density ratios. Thus, the function assumption in our Corollary 1 is close to the function assumption in Zhan et al. 2022, at least in terms of the forms.
> > >
> > > In terms of comparing our Theorem 2 with the work of Zhan et al. 2022, we note that the function assumptions and data assumptions only follow the same principle. There is an inherent gap between the assumptions in these two results because we focus on the lower bound while Zhan et al. considers the upper bound.
> > >
> > > ## Response to Point 3
> > >
> > > Thanks for raising this point.
> > > In section D.2 where the detailed proof of Theorem 1 is provided, we give concrete examples of $\xi$.
> > > In fact, that concrete $\xi(M)$ in section D.2 is optimal w.r.t. $M$ exactly.
> > > On the other hand, we would like to point out that [1] assumes Bellman-completeness, and [2] assumes linear MDPs.
> > > Whether we can learn a better policy comparable with the covered one with only weak function assumptions still remains underexplored.
> > > Addressing this question stands as a major objective in our paper.
> > >
> > > Once again, we appreciate your valuable comments. We are open to receiving any additional comments or questions you may have.
> > >
> > > [1] Bellman-consistent Pessimism for Offline Reinforcement Learning
> > >
> > > [2] is pessimism provably efficient for offline rl

---

> > > > ### Comment · Reviewer_MhTC · 2023-11-21
> > > > **thanks for the follow-up responses**
> > > >
> > > > I have further raised my score to 8 to support the acceptance.

---

### Official Review · Reviewer_REan · 2023-11-02

**Soundness:** 4 excellent
**Presentation:** 4 excellent
**Contribution:** 4 excellent
**Rating:** 8
**Confidence:** 4

**Summary:**

This paper studies offline RL with general function approximation, focusing on an information-theoretic understanding of different types of assumptions (such as completeness and realizability) and how they translate to assumptions on the underlying MDP. This results in a principled way of constructing lower bounds in offline RL with general function approximation via model-realizability, and with this method the authors prove the necessity of completeness assumptions.

**Strengths:**

- This a well-motivated and interesting work and I enjoyed reading the paper. The work puts common assumptions for PAC learning of offline RL with general function approximation under scrutiny and provides a clear overview and concrete definitions of completeness and realizability assumptions.
- Connecting the model-free completeness assumptions to assumptions that impose restrictions on the underlying MDP is very useful as it unifies lower-bound constructions for different function approximation methods. Particularly given that worst-case information-theoretic lower bounds are specific to function-class assumptions—e.g., Foster et al. 2021 prove a lower bound for value-based and exploratory data settings while recent papers such as Zhan et al. 2022 overcame the difficulty by changing the function approximation method.
- With the model-realizability technique introduced in this paper, the authors show the necessity of the completeness-type assumptions.

**Weaknesses:**

There are no major weaknesses.

**Questions:**

--

---

> ### Author Response · Authors · 2023-11-15
>
> We would like to thank the reviewer for the positive assessment of this work. We
> also appreciate your efforts and valuable time for carefully reviewing this paper.

---

### Official Review · Reviewer_85dZ · 2023-11-04

**Soundness:** 3 good
**Presentation:** 3 good
**Contribution:** 3 good
**Rating:** 6
**Confidence:** 3

**Summary:**

This paper studies general function approximations in offline reinforcement learning (RL) settings. The authors firstly formally defined two types of usual assumptions on function approximations, i.e., realizability-type and completeness-type. They then discussed the relationship between assumptions on function approximations and the Markov decision processes (MDPs) those can be realized. Section 5 presented the main Theorem 1, which says that only realizability-type of assumptions are not enough to learn better policies. Corollaries 1-3 then assure similar results for different variants of realizability-type assumptions. The authors later discussed the limitations of the lower bound results, i.e., using over-coverage and not containing $Q^*$-realizability. Theorem 2 then shows a lower bound for $Q^*$-realizability but with partial coverage which may not cover optimal policy.

**Strengths:**

1. The presentation is clear, with enough discussion about the background and related work.
2. The problem is convincingly motivated and important.
3. The results are novel to my knowledge.

**Weaknesses:**

1. As the authors also noted, the main point of arguing that only realizability-type assumptions are not enough to learn better policies is kind of weakened by the fact that upper bounds for $Q^*$-realizability already exist, and Theorem 2 also feels short of compensating since as mentioned the covered policy is not optimal.

**Questions:**

Could you explain how it happens for $\xi(M)$ can be optimal while the value function class does not contain optimal $Q^*$. Example or intuition could be helpful.

---

> ### Author Response · Authors · 2023-11-15
>
> We thank the reviewer for the valuable comment. Here are our responses:
>
> > Could you explain how it happens for $\xi(M)$ can be optimal while the value function class does not contain optimal $Q^\star$. Example or intuition could be helpful.
>
> This is a good question. Note that $\xi(M)$ is only required to be optimal with respect to the initial state distribution,
> whereas $\pi^\star$ should be optimal from every state or state-action pair. The degree of strictness in optimality is the major difference.

---

### Official Review · Reviewer_PRdx · 2023-11-04

**Soundness:** 3 good
**Presentation:** 3 good
**Contribution:** 3 good
**Rating:** 6
**Confidence:** 2

**Summary:**

This paper focuses on offline learning setting with general function approximation.
The authors first formalize different types of assumptions in offline RL (realizability, completeness).
In Section 3, the authors reveal that the lower bounds derived for model-realizability can also be applied to other types of functions.
After that, in Section 4 and 5, they establish lower bounds for learning in offline setting either without $Q^*$-realizability or weaker data coverage assumptions.

**Strengths:**

The paper writing is relatively clear to me. The authors contribute interesting lower bounds, which reveal new understanding for learning in offline setting with general function approximation.

**Weaknesses:**

1. Although lower bound results are also meaningful, it would be better if there are some upper bound results to understand what we can do.

2. As for the results in Section 6, can you explain more about the difference between the data coverage assumptions in (Xie & Jiang 2020) and the lower bound instance in Theorem 2?

    Besides, as for the data coverage assumption used in Theorem 2, how does it compare with the practical scenarios? Would it be too restrictive in practice (maybe in practice we may rarely encounter such bad case)?

**Questions:**

I'm still trying to understand more about Def. 2. Consider the completeness assumption that $\forall f\in\mathcal{F}$ we have $\mathcal{T}^* \in \mathcal{F}$. I wonder what the corresponding $\mathcal{F}$ and $\mathcal{G}$ in Def. 2 should be in this case?

It also seems unclear to me how Def. 2 "captures" what common completeness assumptions ensure: "the existence of a
function class that can minimize a set of loss functions indexed by another function class".

---

> ### Author Response · Authors · 2023-11-14
>
> We thank the reviewer for the valuable comment. Here are our responses:
>
> > Although lower bound results are also meaningful, it would be better if there are some upper bound results to understand what we can do.
>
> The focus and novelty of this work is to develop new lower bounds to understand the fundamental limitation of learnability in offline RL. Several prior works have already established upper bounds on the same problem. In our paper, we have provided a review of these works in Sections 2.2 and 2.3.
>
> > As for the results in Section 6, can you explain more about the difference between the data coverage assumptions in (Xie & Jiang 2020) and the lower bound instance in Theorem 2?
>
> Thanks for your question. In short, the data assumption in section 6 is weaker than the one in (Xie & Jiang 2020)---it is the partial coverage, while the one in (Xie & Jiang 2020) is even more stringent than exploratory coverage.
>
> > Besides, as for the data coverage assumption used in Theorem 2, how does it compare with the practical scenarios? Would it be too restrictive in practice (maybe in practice we may rarely encounter such bad case)?
>
> The data assumption used in Theorem 2 is the partial coverage, which assumes that the dataset covers only one policy. This is practical and is fulfilled in benchmarks like D4RL.
> Furthermore, as we construct information-theoretic lower bounds, we prefer a "restrictive" condition, as it reveals a more crucial fundamental limitation.
>
> > I'm still trying to understand more about Def. 2. Consider the completeness assumption that $\forall f\in\mathcal{F}$, we have $\mathcal{T}^\star\in\mathcal{F}$. I wonder what the corresponding $\mathcal{F}$ and $\mathcal{G}$ in Def. 2 should be in this case?
>
> Bellman-completeness is paired with realizability-type assumptions (e.g., $f^\star\in\mathcal{F}$ where $f^\star$ is the optimal value function) in most cases, thus we can conclude that $\mathcal{F}$ is realizable. Therefore, $\mathcal{F}$ in your question can be considered as $\mathcal{F}$ and $\mathcal{Q}$ in Definition 2 simultaneously. Also, we have that $\mathcal{F}^\star=${$\mathcal{T}^\star f| f\in\mathcal{F}$}.
>
> > It also seems unclear to me how Def. 2 "captures" what common completeness assumptions ensure: "the existence of a function class that can minimize a set of loss functions indexed by another function class".
>
> This is a good question. Note that approximating one target $g\in\mathcal{G}$ with a function class $\mathcal{F}$ is identical to finding $f\in\mathcal{F}$ to minimize the loss $\lVert f-g\rVert_{\infty}$. Thus, the completeness-type assumption essentially imposes a set of loss functions as {$\lVert f-g\rVert_{\infty}|g\in\mathcal{G}$}.

---

### Official Review · Reviewer_XLGY · 2023-11-05

**Soundness:** 3 good
**Presentation:** 2 fair
**Contribution:** 3 good
**Rating:** 6
**Confidence:** 3

**Summary:**

This paper studies the role of function approximation in offline reinforcement learning (RL). Specifically, the paper first propose various types of function classes used in offline RL, and then show the learning lower bound under these types of function classes. Generally speaking, given functions with a policy class domain, it is impossible to learn a good policy.

**Strengths:**

1. The paper proposes a unified way to study general function approximation in offline RL.
2. The theoretical results are solid and interesting.

**Weaknesses:**

1. The presentation is not clear. Some new words, such as exploratory-accurate(accuracy) and data assumption are used without pre-defined.
2. While there are examples about those assumptions, I think a big table summarizing and connecting existing concrete examples and assumptions and the proposed general assumptions can make the paper easier to understand.

**Questions:**

It seems like function approximation on policy class or value function class results in impossible learning tasks, does it mean that learning with a model class might have positive results?

---

> ### Author Response · Authors · 2023-11-14
>
> We thank the reviewer for the valuable comment. Here are our responses:
>
> > 1. The presentation is not clear. Some new words, such as exploratory-accurate(accuracy) and data assumption are used without pre-defined.
>
> Your comments have been well-taken. In fact, the exploratory-accurate is defined right after the definition of the Realizability-type assumption (Definition 1). Different coverage assumptions are defined in the second paragraph in section 2.2.
>
> In the revised version, we have made every effort to provide formal or informal definitions (or at least references to formal definitions) for technical terms when they are first used. For instance, in footnote 2, we provide a reference to the definition places for the terms you point out.
>
> > 2. While there are examples about those assumptions, I think a big table summarizing and connecting existing concrete examples and assumptions and the proposed general assumptions can make the paper easier to understand.
>
> Thanks for your suggestion. In the revision, we have added a table (on page 32 of the Appendices) that compares the assumptions made in this work with those in other related works. We believe this addition further enhances the quality of this paper.
>
> > It seems like function approximation on policy class or value function class results in impossible learning tasks, does it mean that learning with a model class might have positive results?
>
> This is a good question. In fact, both model-free learning and model-based learning have positive results. Whether we have negative or positive results depends on the assumptions of function classes and the data. This paper aims to understand that if we can learn good policies under weak function assumptions in offline RL.

---

> > ### Comment · Reviewer_XLGY · 2023-11-23
> >
> > Thanks for the responses, and I decide to keep my score.

---

### Meta-Review · Area_Chair_WYqu · 2023-12-05

**Metareview:**

The paper addresses the role of function approximation in offline reinforcement learning (RL). It introduces various function classes used in offline RL and establishes learning lower bounds under these classes. The core claim is that learning a good policy is impossible with functions having a policy class domain.

Strengths: The paper's theoretical results are solid and interesting, and the unified framework contributes significantly to the understanding of offline RL with general function approximation.

Weaknesses: Clarity of the paper.

I urge the authors to incorporate the suggestions made by reviewers and those they promised to incorporate for the final version, including: 1) enhancing Appendix F by providing thorough review of the function and data assumptions made in prior works, introducing their definitions and intuitions, and comparing the assumptions in different works in greater detail. 2) expanding Table 4 to offer a clearer comparison among different works.

**Justification For Why Not Higher Score:**

Clarity of the work. If the authors had been willing to make the modifications during the rebuttal phase promised, I maybe would have considered an oral.

**Justification For Why Not Lower Score:**

The authors sufficiently clarified concerns during the rebuttal phase in the responses, and the contribution is relevant and significant.

---

### Decision · Program_Chairs · 2024-01-16

Accept (spotlight)